# Data Debugging with Shapley Importance over Machine Learning Pipelines

**Bojan Karlaš[1]\*, David Dao[2], Matteo Interlandi[3], Sebastian Schelter[4], Wentao Wu[3], Ce Zhang[5]**
[1]Harvard University, [2]ETH Zurich, [3]Microsoft, [4]University of Amsterdam, [5]University of Chicago
\*bkarlas@mgh.harvard.edu

## Abstract

When a machine learning (ML) model exhibits poor quality (e.g., poor accuracy or fairness), the problem can often be traced back to errors in the training data. Being able to discover the data examples that are the most likely culprits is a fundamental concern that has received a lot of attention recently. One prominent way to measure "data importance" with respect to model quality is the Shapley value. Unfortunately, existing methods only focus on the ML model in isolation, without considering the broader ML pipeline for data preparation and feature extraction, which appears in the majority of real-world ML code. This presents a major limitation to applying existing methods in practical settings. In this paper, we propose Datascope, a method for efficiently computing Shapley-based data importance over ML pipelines. We introduce several approximations that lead to dramatic improvements in terms of computational speed. Finally, our experimental evaluation demonstrates that our methods are capable of data error discovery that is as effective as existing Monte Carlo baselines, and in some cases even outperform them. We release our code as an open-source data debugging library available at github.com/easeml/datascope.

## 1 Introduction

Data quality issues have been widely recognized to be among the main culprits for underperforming machine learning (ML) models, especially when it comes to tasks that are otherwise considered solved by ML (Liang et al., 2022; Ilyas & Chu, 2019). A common type of data errors are wrong labels. For example, biomedical images can be misdiagnosed due to human error which results in label errors. Many systematic methods have been developed to repair *data errors* (Rekatsinas et al., 2017; Krishnan et al., 2017). Unfortunately, in many practical scenarios, repairing data in a reliable manner requires human labor, especially if humans have been involved in producing the original data. The high cost of this *data debugging* process has led to a natural question – Can we prioritize data repairs based on some notion of *importance* which leads to the highest quality improvements for the downstream model?

In recent years, several approaches have emerged to answer these questions. One line of work suggests expressing importance using *influence functions* (Koh & Liang, 2017) which is essentially a gradient-based approximation of the leave-one-out (LOO) method. Here, the importance of a training data example is expressed as the difference in the model quality score observed after removing that data example from the training set. This quality difference is referred to as the *marginal contribution* of that data example. Another line of work proposes *Shapley value* as a measure of importance (Ghorbani & Zou, 2019; Jia et al., 2019b; 2021) that has a long history in game theory (Shapley, 1951). In the context of data importance, it can be seen as a generalization of LOO. Namely, instead of measuring the marginal contribution over the entire training set, we measure it over every subset of the training set and then compute a weighted average. Apart from having many useful theoretical properties, the Shapley value was shown to be very effective in many data debugging scenarios (Jia et al., 2021).

On the flip side, because the Shapley value requires enumerating *exponentially* many subsets, computing it is *intractable* in practical settings. There have been different ways to *approximate* this computation. This includes Monte Carlo (MC) sampling (Ghorbani & Zou, 2019) or group testing (Jia et al., 2019b) to sample subsets of training data, train models as black boxes on those subsets, compute marginal contributions of training data examples, and aggregate the results to compute the final approximated result. Unfortunately, re-training the model can be quite costly, especially for large models. Some methods try to overcome this by leveraging proxy models such as K-nearest neighbors

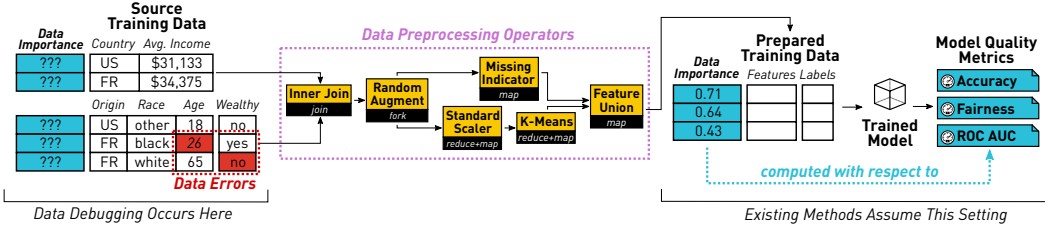

Figure 1: Existing data debugging methods were designed to compute data importance of already preprocessed data. In typical real-world scenarios, data errors occur in source datasets, before being passed through a data preparation pipeline. The goal of our work is to help close that gap.

(KNN) (Jia et al., 2019a) and exploiting its simple structure to derive dynamic programming (DP) algorithms for computing the Shapley value.

One major trait of the existing work in this space is that it primarily focuses on computing the importance of data examples in the *prepared training dataset*. This poses a challenge in practical settings where data errors typically occur earlier in the data preparation process. In most realistic scenarios, this process involves taking one or more *source training datasets*, joining them together if needed, and applying a composition of data preprocessing operators (Figure 1). The simplest operators may represent a 1-1 *map* of input dataset elements to output dataset elements (referred to as *tuples* in the data management literature). Some operators, such as the data augmentation operator, *fork* the data by converting a single input data tuple into multiple output tuples. On the other hand, an output tuple of a *join* operator can be the product of multiple input tuples. Finally, some operators involve a *reduce* operation which involves computing some intermediate result based on the entire input dataset (e.g. the mean and standard deviation) and then applying that result to output tuples.

This new setting impacts existing approximation methods in several ways. Firstly, given that it is a black box approach, Monte Carlo sampling (Ghorbani & Zou, 2019) can directly be applied to this setting. However, this comes with the computational cost of re-running the data preprocessing operators for every subset of the training data that we sample. Depending on the complexity of the preprocessing pipeline, this cost can be quite significant. Secondly, the existing KNN-based Shapley approximation method (Jia et al., 2019a) strictly relies on the ability to independently remove tuples from the prepared training dataset in order to compute their marginal contributions. Given the aforementioned complexity induced by preprocessing operators, the tractability result of the previous KNN-based method does not hold directly in this new setting. Therefore, a novel analysis is needed to see whether the Shapley value computation can be made tractable depending on the structure of the data preprocessing pipeline.

**Contributions.** In this paper, we focus on studying the relationship between the structure of ML pipelines and our ability to efficiently compute Shapley values of source data examples. We make use of data provenance (Green et al., 2007; Cheney et al., 2009), a simple yet powerful theoretical toolkit for tracing individual data examples as they pass through a data processing pipeline. We propose ease.ml/datascope, a framework for modeling the interdependence between tuples induced by data preprocessing operators. Our contributions can be summarized as follows:

- We apply the notion of data provenance to ML pipelines in order to relate the input and output datasets as a function of the pipeline structure. We introduce the notion of a *"canonical pipeline"* which we simply define as a distinct pipeline structure that lends itself to efficiently relating pipeline inputs and outputs, as well as efficiently computing Shapley values. We identify three classes of canonical pipelines: map, fork and one-to-many join. (section 3)
- We show how approximating pipelines as canonical leads to significant speed-ups of Monte Carlo methods for Shapley computation. We also demonstrate how the majority of real-world ML pipelines can be approximated as canonical. (section 3)
- We combine canonical pipelines with the K-nearest neighbor as a proxy model. We show how canonical pipelines can be compiled into efficient counting oracles and used to derive PTIME Shapley computation algorithms. Under this framework, the KNN Shapley method from prior work represents a special case applicable to map pipelines. (section 4)
- We conduct an extensive experimental evaluation by applying all considered Shapley computation methods to the task of repairing noisy labels in various real-world datasets. We conclude that in most cases our method is able to achieve solid performance in terms of reducing the cost of label repair while demonstrating significant improvements in computational runtime. (section 5)

## 2  PROBLEM: COMPUTING THE SHAPLEY VALUE OVER ML PIPELINES

**Shapley Value.** Let $\mathcal{D}_{tr}$ be a training dataset and $u$ some utility function used to express the *value* of any subset of $\mathcal{D}_{tr}$ by mapping it to a real number. Then, the Shapley value, denoting the importance of a tuple $t_i \in \mathcal{D}_{tr}$, is defined as

$$\varphi(t_i) = \frac{1}{|\mathcal{D}_{tr}|} \sum_{\mathcal{D} \subseteq \mathcal{D}_{tr} \setminus \{t_i\}} \binom{|\mathcal{D}_{tr}|-1}{|\mathcal{D}|}^{-1} \left( u(\mathcal{D} \cup \{t_i\}) - u(\mathcal{D}) \right). \tag{1}$$

Intuitively, the *importance* of $t_i$ for a subset $\mathcal{D} \subseteq \mathcal{D}_{tr} \setminus \{t_i\}$ is measured as the difference between the utility $u(\mathcal{D} \cup \{t_i\})$ *with* $t_i$ and the utility $u(\mathcal{D})$ *without* $t_i$. The Shapley value takes a weighted average of all of the $2^{|\mathcal{D}_{tr}|-1}$ possible subsets $\mathcal{D} \subseteq \mathcal{D}_{tr} \setminus \{t_i\}$, which enables it to have a range of desired properties that significantly benefit data debugging tasks, often leading to more effective data debugging mechanisms compared to other leave-one-out methods.

**Quality of ML Pipelines.** As mentioned, the utility function $u$ is defined to measure the value of any subset of $\mathcal{D}_{tr}$, which in our context corresponds to the *source training dataset*. We assume that this dataset can be made up of multiple sets of tuples (e.g. a multi-modal dataset involving a set of images and a table with metadata). The *validation dataset* $\mathcal{D}_{val}$ is defined in a similar manner.

Then, let $f$ be a *data preprocessing pipeline* that transforms any training data subset $\mathcal{D} \subseteq \mathcal{D}_{tr}$ into a set of tuples $\{t_i = (x_i, y_i)\}_{i \in [M]}$ made up of $M$ feature and label pairs that the ML training algorithm $\mathcal{A}$ takes as input. Finally, we obtain a trained ML model $\mathcal{A} \circ f(\mathcal{D})$ which we can evaluate using some model quality metric. Based on this, we can define the utility function $u$ used to express the value of a training data subset $\mathcal{D}$ as a measure of the quality of an ML pipeline $\mathcal{A} \circ f(\mathcal{D})$ when scored using $\mathcal{D}_{val}$. Formally, we write this as

$$u(\mathcal{D}) := m(\mathcal{A} \circ f(\mathcal{D}), f(\mathcal{D}_{val})). \tag{2}$$

Here, $m$ can be any model quality metric such as accuracy or a fairness metric such as equalized odds difference. Note that, for simplicity, we assume that we are applying the same pipeline to both the training data subset $\mathcal{D}$ and the validation dataset $\mathcal{D}_{val}$. In general, these two pipelines can differ as long as the data format of $f(\mathcal{D}_{val})$ is readable by the trained ML model. For example, a data augmentation operation is typically applied to training data only (as is the case in our experiments).

**Core Technical Problem.** In this work, we focus on the ML pipeline utility $u$ defined in Equation 2 and we ask the following question: *How can we approximate the structure of $u$ in order to obtain Shapley-based data importance that is (1) computationally fast; and (2) effective at downstream data debugging tasks?*

## 3  CANONICAL ML PIPELINES

In this section, we take a closer look at a data preprocessing pipeline $f$ that can, in principle, contain an arbitrarily complex set of data processing operators. This complexity can result in a heavy overhead on the cost of computing the Shapley value. This overhead comes from having to re-evaluate the pipeline many times for different training data subsets. In this section, we describe a framework for minimizing that overhead by solving a concrete technical problem.

**Problem 1.** We are given a training dataset $\mathcal{D}_{tr}$, a data preprocessing pipeline $f$, and the output set $f(\mathcal{D}_{tr})$. For an arbitrary subset $\mathcal{D} \subseteq \mathcal{D}_{tr}$ and some tuple $t' \in f(\mathcal{D}_{tr})$, decide whether $t' \in f(\mathcal{D})$ in time $O(1)$ w.r.t. $|\mathcal{D}_{tr}|$.

It is easy to see how solving this problem virtually removes the cost of computing the pipeline output of an arbitrary training data subset. Next, we describe a reduced version of the *data provenance* framework (Green et al., 2007; Cheney et al., 2009) which we will apply to solve this problem.

### 3.1  DATA PROVENANCE FOR ML PIPELINES

We define a set of binary variables $A$ and associate a variable $a_t \in A$ with every training data tuple $t \in \mathcal{D}_{tr}$. Each subset $\mathcal{D} \subseteq \mathcal{D}_{tr}$ can be defined using a *value assignment* $v(a) \mapsto \{0, 1\}$, where $v(a_t) = 1$ means that $t \in \mathcal{D}$. We can use $\mathcal{D}_{tr}[v]$ to denote $\mathcal{D}$. We write $\mathcal{V}_A$ to denote the set of all the $2^{|A|}$ possible value assignments. Next, with every tuple $t' \in f(\mathcal{D}_{tr})$ we associate a "provenance polynomial" $p_{t'}$ which is a logical formula with variables in $A$ (e.g. $a_1 + a_2 \cdot a_3$). For a given value assignment $v$, we define an evaluation function $\text{eval}_v(p_{t'}) \mapsto \{0, 1\}$ which simply follows the

(a) Map pipeline     (b) Fork pipeline     (c) One-to-many join pipeline     (d) Distribution of canonical pipelines

Figure 2: (a-c) Three types of canonical pipelines where data provenance allows us to efficiently compute subsets. (d) A majority of real-world ML pipelines (Psallidas et al., 2019) either already exhibit a canonical pipeline pattern, or are easily convertible to it using our approximation scheme.

standard logical reduction rules to determine the truthiness of $p_{t'}$ given $v$. For a tuple $t' \in f(\mathcal{D}_{tr})$ and a value assignment $v$, we define $t' \in f(\mathcal{D}_{tr}[v])$ iff $\mathrm{eval}_v(p_{t'}) = 1$. It is easy to see that we can directly apply this framework to solve Problem 1. However, to respect the $O(1)$ time complexity, $|p_t|$ must be $O(1)$ w.r.t. $|\mathcal{D}_{tr}|$. In subsection 3.2, we explore when this condition is met.

**Redefining the Shapley value.** Using this framework, we can rewrite the Shapley value as:

$$\varphi(t_i) = \frac{1}{|A|} \sum_{v \in \mathcal{V}_{A \setminus \{a_i\}}} \binom{|A|-1}{|\mathrm{supp}(v)|}^{-1} u(\mathcal{D}_{tr}[v; a_i \leftarrow 1]) - u(\mathcal{D}_{tr}[v; a_i \leftarrow 0]) \qquad (3)$$

The notation $[v; a_i \leftarrow X]$ for $X \in \{0, 1\}$ means that we augment $v$ with $v(a_i) = X$. Also, we define the support of $v$ as $\mathrm{supp}(v) := \{a \in A : v(a) = 1\}$.

## 3.2 Approximation: ML Pipelines are Canonical

As mentioned above, solving Problem 1 in $O(1)$ time depends on $|p_t|$ being $O(1)$ w.r.t. $|\mathcal{D}_{tr}|$. This does not necessarily hold true for an arbitrary pipeline $f$. However, it does hold true for some classes of pipelines, which we refer to as *canonical pipelines*. Hence, if we approximate the pipeline $f$ as canonical, then we can solve Problem 1. The three classes of canonical pipelines that we identified to be useful in the context of this work are: map, fork, and one-to-many join pipelines (Figure 2).

**Map pipelines.** This is the simplest form of pipeline where each input tuple $t \in \mathcal{D}_{tr}$ corresponds to at most one output tuple $t' \in f(\mathcal{D}_{tr})$, after passing through an optional per-tuple mapping function $\mu(t) \mapsto t'$ (Figure 2a). Examples of such pipelines include missing value indicators, polynomial feature generators, pre-trained embeddings, etc.

**Fork pipelines.** In this pipeline, each input tuple $t \in \mathcal{D}_{tr}$ can be associated with multiple output tuples $t' \in f(\mathcal{D}_{tr})$, but a single output tuple is associated with a single input tuple (Figure 2b). A prominent example is a data augmentation pipeline that outputs several slightly altered versions of every input tuple.

**One-to-many Join pipelines.** This pipeline contains table join operators like the one in Figure 1. Here, the training dataset $\mathcal{D}_{tr} = \{\mathcal{D}_t, \mathcal{D}_{a_1}, \ldots, \mathcal{D}_{a_k}\}$ is made up of multiple tuple sets that form a "star schema". This means that any training example tuple $t \in \mathcal{D}_t$ can be joined with no more than one tuple from each of the auxiliary tables $\mathcal{D}_{a_1}, \ldots, \mathcal{D}_{a_k}$. Note that, for this pipeline, the provenance polynomial of each output tuple is a Boolean product of variables associated with all tuples that were joined to produce that output tuple (Figure 2c).

## 3.3 Approximating Real ML Pipelines

Even though many real-world pipelines can be directly represented as our canonical pipelines, there is still a solid amount that cannot be represented. Nevertheless, upon taking a closer look, we can identify a class of pipelines that we might be able to approximately represent. These are the pipelines that exhibit an *estimator-transformer* pattern $f(\mathcal{D}) = \mathrm{map}(\mathrm{reduce}(\mathcal{D}), \mathcal{D})$. Specifically, they are made up of some reduce operation performed on the entire dataset which produces some intermediate data, which is used to parameterize a map operation which is performed on individual tuples. An example of such a pipeline is a min-max scaler, where the reduce step computes min and max statistics for each feature, which are then used to re-scale individual tuples.

The reduce step of this pipeline causes every output tuple to depend on every input tuple, which does not fit into our canonical pipeline framework. However, we can still approximate such pipelines by isolating the intermediate data produced by $\mathrm{reduce}(\mathcal{D}_{tr})$. Then, conditioned on that intermediate data, we can re-define our pipeline $f$ to be a *conditional map* pipeline $f^*$ as follows:

$$f(\mathcal{D}) = \mathrm{map}(\mathrm{reduce}(\mathcal{D}), \mathcal{D}) \mapsto f^*(\mathcal{D}) = \mathrm{map}(\mathrm{reduce}(\mathcal{D}_{tr}), \mathcal{D}).$$

**Evaluation of Effectiveness.** We evaluate our method of approximating pipelines as canonical and apply it directly to compute the Shapley value using the Truncated Monte Carlo (TMC) sampling

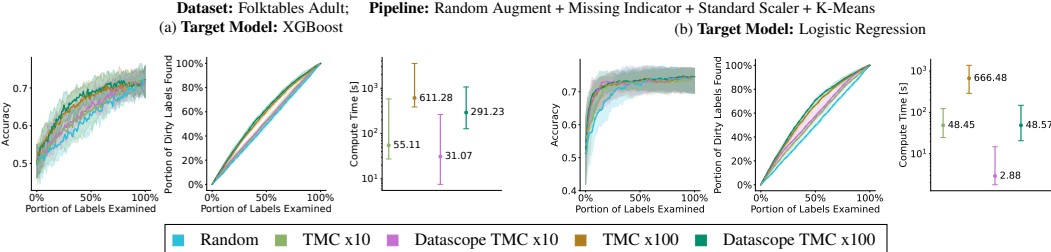

Figure 3: An ML pipeline with an *estimator-transformer* pattern approximated as a canonical pipeline can achieve comparable performance on a label repair task, with significantly faster runtime.

method (Ghorbani & Zou, 2019). We run the evaluation for 10 and 100 Monte Carlo iterations (x10/x100). We can see that our approach exhibits comparable performance with significant gains in computational runtime (Figure 3). See section 5 for more details about the experimental protocol.

**Statistics of Real-world Pipelines.** A natural question is how common these families of pipelines are in practice. Figure 2d illustrates a case study that we conducted using 500K real-world pipelines provided by Microsoft (Psallidas et al., 2019). We divide pipelines into three categories: (1) "pure" map/fork pipelines, based on our definition of canonical pipelines; (2) "conditional" map/fork pipelines, which are comprised of a reduce operator that can be effectively approximated using the scheme we just described; and (3) other pipelines, which contain complex operators that cannot be approximated. We observe that a vast majority of pipelines we encountered in our case study fall into the first two categories that we can effectively approximate using our canonical pipelines framework.

## 4 SHAPLEY VALUE OVER CANONICAL PIPELINES

In section 3 we described an approach for treating the data preprocessing pipeline $f$ as a white box which led us to directly attainable performance improvements of Monte Carlo Shapley methods. However, these methods still rely on treating the model $\mathcal{A}$ as a black box and retraining it for different training data subsets, which often results in slow runtime. In this section, we are interested in PTIME algorithms that give orders of magnitude faster runtime and thus open the door for interactive data debugging. Specifically, we focus on the following technical problem:

**Problem 2.** We are given a training dataset $\mathcal{D}_{tr}$, a data preprocessing pipeline $f$ and a model quality metric $m$ computed over a given validation dataset $\mathcal{D}_{val}$. Compute the Shapley value (as defined in Equation 1) of a given tuple $t_i \in \mathcal{D}_{tr}$ for the ML pipeline utility (as defined in Equation 2) in time polynomial w.r.t. $|\mathcal{D}_{tr}|$ and $|\mathcal{D}_{val}|$.

We will now explore additional approximations we can make on the model $\mathcal{A}$ as well as the model quality metric $m$. Specifically, we replace the model with a KNN classifier, and we assume that the quality metric has a specific additive structure. We then sketch the outline of a solution to the given problem that leverages these approximations. It should be noted that although prior work has explored the idea of using the KNN proxy model for PTIME algorithms (Jia et al., 2019a), to the best of our knowledge, the work presented in this paper is the first to analyze the relationship between the structure of different types of ML pipelines and the computational complexity of the Shapley value computation. A brief discussion about the limitations of prior work is presented in Appendix A.

### 4.1 APPROXIMATION: THE MODEL IS KNN AND THE MODEL QUALITY METRIC IS ADDITIVE

Here we define two structures which we will use as building blocks for approximating ML pipelines: the KNN model and additive model quality metrics. In the following section we will show how these building blocks can be leveraged to provide PTIME algorithms for computing Shapley values.

**K-Nearest Neighbor (KNN) Model.** We provide a specific definition of the KNN model in order to facilitate our further analysis. Given some set of training tuples $\mathcal{D}$ and a validation tuple $t_{val}$, the KNN model $\mathcal{A}_{KNN}(\mathcal{D})$ can be defined as follows:

$$\mathcal{A}_{KNN}(\mathcal{D})(t_{val}) := \text{argmax}_{y \in \mathcal{Y}}\bigg(\text{tally}\Big(\mathcal{D} \mid \text{top}_K\big(\mathcal{D} \mid t_{val}\big), t_{val}\Big)(y)\bigg). \quad (4)$$

Here, $\text{top}_K(\mathcal{D} \mid t_{val})$ returns a tuple $t_K \in \mathcal{D}$ that takes the $K$-th position when ranked by similarity with the validation tuple $t_{val}$. Furthermore, $\text{tally}(\mathcal{D} \mid t_K, t_{val})$ tallies up the class labels of all tuples

in $\mathcal{D}$ that have similarity with $t_{val}$ higher or equal to $t_K$. It returns $\gamma$, a label tally vector that is indexed by class labels (i.e. $\gamma : \mathcal{Y} \to \mathbb{N}$). Note that the sum of all elements in $\gamma$ must be $K$. Given a set of classes $\mathcal{Y}$, we define $\Gamma_{\mathcal{Y},K}$ to be the set of all possible label tally vectors. Finally, assuming a standard majority voting scheme, $\operatorname{argmax}_{y \in \mathcal{Y}}$ returns the predicted class label with the highest tally.

**Additive Model Quality Metric.** We say that a model quality metric is *additive* if there exists a tuple-wise metric $m_T$ such that $m$ can be written as:

$$m(\mathcal{A} \circ f(\mathcal{D}), f(\mathcal{D}_{val})) = w \cdot \sum_{t_{val} \in f(\mathcal{D}_{val})} m_T\left(\Big(\mathcal{A} \circ f(\mathcal{D})\Big)(t_{val}), t_{val}\right) \tag{5}$$

Here, $w$ is a scaling factor that can depend only on $\mathcal{D}_{val}$. The tuple-wise metric $m_T : (y_{pred}, t_{val}) \mapsto [0, 1]$ takes a validation tuple $t_{val} \in \mathcal{D}_{val}$ as well as a class label $y_{pred} \in \mathcal{Y}$ predicted by the model for $t_{val}$. It is easy to see that some popular utilities, such as validation accuracy, are additive, e.g., the accuracy utility is simply defined by plugging $m_T(y_{pred}, (x_{val}, y_{val})) := \mathbb{1}\{y_{pred} = y_{val}\}$ and $w := 1/|\mathcal{D}_{val}|$ into Equation 5. In subsection E.3, we show even more examples of such metrics.

## 4.2 Computing the Shapley Value

We now outline our approach to computing the Shapley value of a training data tuple $t_i \in \mathcal{D}_{tr}$ using our approximation described in subsection 3.2 and subsection 4.1. We start off from Equation 3 and plug in $u$ as defined in Equation 2. Next, since we assume that our model quality metric is additive, we plug in $m$ as defined in Equation 5. By rearranging the sums, we can write the Shapley formula as $\varphi(t_i) = w \cdot \sum_{t_{val} \in f(\mathcal{D}_{val})} \varphi(t_i, t_{val})$, where $\varphi(t_i, t_{val})$ is a validation tuple-wise Shapley value. Under the assumption that our model is KNN, we can plug in $\mathcal{A}$ as defined in Equation 4, rearrange the sums, and arrive at the following definition of $\varphi(t_i, t_{val})$:

$$\varphi(t_i, t_{val}) = \frac{1}{|A|} \sum_{t', t'' \in f(\mathcal{D}_{tr})} \sum_{\alpha=1}^{|A|} \binom{|A|-1}{\alpha}^{-1} \sum_{\gamma', \gamma'' \in \Gamma_{\mathcal{Y},K}} m_\Delta(\gamma', \gamma'' \mid t_{val}) \cdot \omega(\alpha, \gamma', \gamma'' \mid t_i, t_{val}, t', t''). \tag{6}$$

We define $m_\Delta(\gamma', \gamma'' \mid t_{val}) := m_T(\operatorname{argmax}_{y \in \mathcal{Y}} \gamma''(y), t_{val}) - m_T(\operatorname{argmax}_{y \in \mathcal{Y}} \gamma'(y), t_{val})$ as the differential metric.

**Counting Oracles.** The function $\omega$ in Equation 6 is a *counting oracle* which we introduce to help us isolate and analyze the exponential sum from Equation 3. We define it as:

$$\omega(\alpha, \gamma', \gamma'' \mid t_i, t_{val}, t', t'') := \sum_{v \in \mathcal{V}_{A \setminus \{a_i\}}} \cdot \mathbb{1}\Big\{\alpha = |\operatorname{supp}(v)|\Big\}$$
$$\cdot \mathbb{1}\Big\{t' = \operatorname{top}_K\big(f(\mathcal{D}_{tr}[v; a_i \leftarrow 0]) \mid t_{val}\big)\Big\} \cdot \mathbb{1}\Big\{t'' = \operatorname{top}_K\big(f(\mathcal{D}_{tr}[v; a_i \leftarrow 1]) \mid t_{val}\big)\Big\} \tag{7}$$
$$\cdot \mathbb{1}\Big\{\gamma' = \operatorname{tally}\big(f(\mathcal{D}_{tr}[v; a_i \leftarrow 0]) \mid t', t_{val}\big)\Big\} \cdot \mathbb{1}\Big\{\gamma'' = \operatorname{tally}\big(f(\mathcal{D}_{tr}[v; a_i \leftarrow 1]) \mid t'', t_{val}\big)\Big\}.$$

Intuitively, the counting oracle is a function that returns the number of value assignments with exactly $\alpha$ variables set to 1, and the label tally of the top-K tuples will be exactly $\gamma''$ when $t_i$ is included in the training dataset, and $\gamma'$ when it is excluded. By looking at Equation 6 we can observe that all the sums are polynomial w.r.t. the size of data. Thus, we arrive at the following theorem (which we prove in Appendix E):

**Theorem 4.1.** *If we can compute the counting oracle $\omega$ as defined in Equation 7 in time polynomial w.r.t. $|\mathcal{D}_{tr}|$ and $|\mathcal{D}_{val}|$, then we can compute the Shapley value of a tuple $t_i \in \mathcal{D}_{tr}$ in time polynomial w.r.t. $|\mathcal{D}_{tr}|$ and $|\mathcal{D}_{val}|$.*

The above theorem outlines a solution of Problem 2, given that we can find a PTIME solution for computing the counting oracle. Next, we cover a solution that models the problem as a model counting problem by leveraging a data structure which we call Additive Decision Diagrams (ADD's).

**Counting Oracle as Model Counting over ADD's.** We use Additive Decision Diagram (ADD) to compute the counting oracle $\omega_{t,t'}$ (Equation 7). An ADD represents a Boolean function $\phi : \mathcal{V}_A \to \mathcal{E} \cup \{\infty\}$ that maps value assignments $v \in \mathcal{V}_A$ to elements of some set $\mathcal{E}$ or a special invalid element $\infty$ (see Appendix C for more details). For our purpose, we define $\mathcal{E} := \{1, ..., |A|\} \times \Gamma_{\mathcal{Y},K} \times \Gamma_{\mathcal{Y},K}$. Then, we define a function over Boolean inputs $\phi(v \mid t_i, , t_{val}, t', t'')$ as follows:

$$\phi(v \mid t_i, t_{val}, t', t'') := \begin{cases} \infty, & \text{if } t' \notin \mathcal{D}_{tr}[v; a_i \leftarrow 0], \\ \infty, & \text{if } t'' \notin \mathcal{D}_{tr}[v; a_i \leftarrow 1], \\ (\alpha, \gamma', \gamma''), & \text{otherwise}, \end{cases} \tag{8}$$
$$\alpha := |\operatorname{supp}(v)|, \quad \gamma' := \operatorname{tally}\big(\mathcal{D}_{tr}[v; a_i \leftarrow 0] \mid t', t_{val}\big), \quad \gamma'' := \operatorname{tally}\big(\mathcal{D}_{tr}[v; a_i \leftarrow 1] \mid t'', t_{val}\big).$$

Figure 4: Computing the Shapley value by using KNN as a proxy model can achieve comparable performance on a label repair task, with orders of magnitude faster runtime.

If we can construct an ADD that computes $\phi(v \mid t_i, t_{val}, t', t'')$, then the model counting operation on that ADD exactly computes $\omega(\alpha, \gamma', \gamma'' \mid t_i, t_{val}, t', t'')$. As the complexity of model counting is $O(|\mathcal{N}| \cdot |\mathcal{E}|)$ (see Equation 12) and $|\mathcal{E}|$ is polynomial in the data size, we have the following result:

**Theorem 4.2.** *If we can represent the $\phi_{t,t'}(v)$ in Equation 8 with an ADD of size polynomial in $|A|$ and $|\mathcal{D}_{tr}^f|$, we can compute the counting oracle $\omega_{t,t'}$ in time polynomial of $|A|$ and $|\mathcal{D}_{tr}^f|$.*

A proof is provided in Appendix E. For specific canonical pipelines, we have the following corollaries.

**Corollary 4.1. (One-to-Many Join Pipelines)** *For the $K$-NN accuracy utility and a one-to-many* join *pipeline, which takes as input two datasets, $\mathcal{D}_F$ and $\mathcal{D}_D$, of total size $|\mathcal{D}_F| + |\mathcal{D}_D| = N$ and outputs a joined dataset of size $O(N)$, the Shapley value can be computed in $O(N^4)$ time.*

**Corollary 4.2. (Fork Pipelines)** *For the $K$-NN accuracy utility and a* fork *pipeline, which takes as input a dataset of size $N$ and outputs a dataset of size $M$, the Shapley value can be computed in $O(M^2N^2)$ time.*

**Corollary 4.3. (Map Pipelines)** *For the $K$-NN accuracy utility and a* map *pipeline, which takes as input a dataset of size $N$, the Shapley value can be computed in $O(N^2)$ time.*

**Evaluation of Effectiveness.** We evaluate our method of computing the Shapley value by using KNN as a proxy model (Figure 4). We can see that its effectiveness is comparable even when applied to the task of label repair over pipelines that have different models. On the other hand, we can see that the computational cost is orders of magnitude lower when compared to MC methods.

## 5 EXPERIMENTAL EVALUATION

We evaluate the performance of our method by applying it to a common data debugging scenario – label repair. The goal of this empirical study was to validate that: (1) our approximations enable significantly faster computation of Shapley values, and (2) in spite of any inherent biases, these approximations still manage to enable effective data debugging.

### 5.1 EXPERIMENTAL SETUP

**Protocol.** We conduct a series of experimental runs that simulate a real-world importance-driven data debugging workflow. We developed a custom experimental infrastructure based on dcbench (Eyuboglu et al., 2022). In each experimental run, we select a dataset, pipeline, model, and data repair method. If a dataset does not already have human-generated label errors, we follow the protocol of Li et al. (2021) and Jia et al. (2021) and artificially inject $50\%$ of label noise. We compute the importance using a *validation dataset* and use it to prioritize our label repairs. We divide the range between $0\%$ data examined and $100\%$ data examined into 100 checkpoints. At each checkpoint, we measure the quality of the given model on a separate *test dataset* using some metric (e.g. accuracy). We also measure the time spent on computing importance scores for the entire training dataset. We repeat each experiment 10 times and report the median as well as the 90-th percentile range (either shaded or with error bars).

**Data Debugging Methods.** We apply various methods of computing data importance:

- Random — Importance is a random number and thus we apply data repairs in random order.
- TMC x10 / x100 — Shapley values computed using the Truncated Monte-Carlo (TMC) method (Ghorbani & Zou, 2019), with 10 and 100 Monte-Carlo iterations, respectively.
- Datascope TMC x10 / x100 — This applies our method of approximating pipelines using data provenance over canonical pipelines to the TMC method of computing the Shapley value.

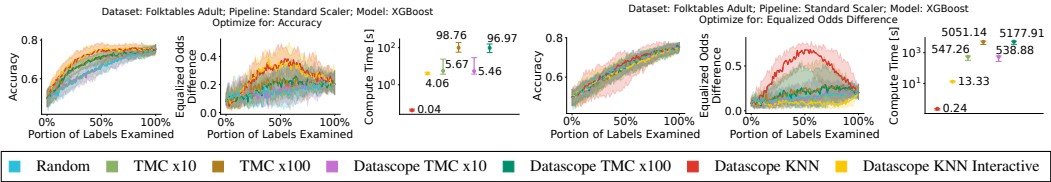

Figure 5: Under our framework it is possible to optimize for model quality metrics other than accuracy. Here we show a commonly used fairness metric – equalized odds difference (lower is better). Given that approximating this metric is more complex, optimal results are achieved by using KNN Interactive which recomputes the Shapley value after each data repair checkpoint.

- Datascope KNN — This is our method for efficiently computing the Shapley value over ML pipelines by using the KNN as a proxy model.
- Datascope KNN Interactive — While the above methods compute importance only once at the beginning of the repair process, the speed of our method allows us to *recompute* the importance after *each* data repair checkpoint.

**Data Preprocessing Pipelines.** We obtained a dataset with about $500K$ machine learning workflow instances from internal Microsoft users (Psallidas et al., 2019). Each workflow consists of a dataset, a data preprocessing pipeline, and an ML model. We identified a handful of the most representative pipelines and translated them to `sklearn` pipelines. All pipelines used in our experiments are listed in Table 1 along with the operators they are made up of. Some pipelines are purely canonical, while some involve a reduce operation.

Table 1: Data preprocessing pipelines used in experiments.

| Pipeline | Dataset Modality | Purely Canonical | Operators |
|---|---|---|---|
| Identity | tabular | true | ∅ |
| Standard Scaler | tabular | false | StandardScaler |
| Logarithmic Scaler | tabular | false | Log1P ∘ StandardScaler |
| PCA | tabular | false | PCA |
| Missing Indicator + KMeans | tabular | false | MissingIndicator ∘ KMeans |
| Gaussian Blur | image | true | GaussBlur |
| Histogram of Oriented Gradients | image | true | HogTransform |
| ResNet18 Embedding Model | image | true | ResNet18 |
| MobileViT Embedding (Mehta & Rastegari, 2022) | image | true | MobileViT |
| TFIDF | text | false | CountVectorizer ∘TfidfTransformer |
| Tolower + URLRemove + TFIDF | text | false | TextToLower ∘ UrlRemover ∘CountVectorizer ∘TfidfTransformer |
| MinLM Embedding (Wang et al., 2020) | text | true | MinLM |
| ALBERT Embedding (Reimers & Gurevych, 2019) | text | true | Albert |

## 5.2 RESULTS

In this section, we highlight some of the most interesting results of our empirical analysis and point out some key insights that we can draw. A more extensive experimental analysis is presented in Appendix G. We start off with three general scenarios: (1) accuracy-driven label repair; (2) fairness-driven label repair to demonstrate usage of different model quality metrics; and (3) label repair in deep learning scenarios. In each one, we study the tradeoff between *computational cost* of any data repair approach, and the *labor cost*, which is measured as the amount of data repairs that need to be conducted to deliver the biggest improvement of model quality. Finally, we conduct a scalability analysis of our algorithm to showcase its potential for handling large datasets.

**Improving Accuracy.** In this set of experiments, our goal is to improve model accuracy with targeted label repairs. In Figure 4 we show one example workflow for the FolkUCI Adult dataset and the pipeline from Figure 1 without the join operator. We evaluate our KNN-based method over pipelines that contain two different ML models: LogisticRegression and XGBoost. We can draw two key conclusions about our KNN-based algorithm. Firstly, given that our KNN-based method is able to achieve comparable performance to Monte Carlo-based methods, we can conclude that KNN can indeed serve as a good proxy model for computing the Shapley value. Secondly, it is able to achieve this performance at only a fraction of the computational cost which makes it even more compelling.

**Improving Accuracy and Fairness.** Next, we explore the relationship between accuracy and fairness when performing label repairs. In these experiments, we use tabular datasets that have a 'sex' feature that we use to compute group fairness using *equalized odds difference* (Hardt et al., 2016). In Figure 5 we explore the tradeoff between two data debugging goals – the left panel illustrates the behavior of optimizing for accuracy whereas the right panel illustrates the behavior of optimizing for fairness. We first notice that being able to debug specifically for fairness is important because for some datasets improving accuracy does not necessarily improve the fairness of the trained model. Secondly, we can see that even when we do optimize for fairness, not all methods will end up being successful. The best-performing method is Datascope KNN Interactive which is the only one that recomputes the Shapley value at each of the 100 checkpoints (due to the speed of our KNN-based method). It is likely that the complexity of the equalized odds difference as a metric makes it challenging to

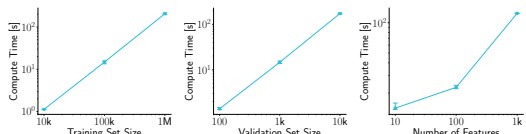

(a) ResNet as target Model
(b) ResNet embedding as the preprocessing pipeline
(c) Matching Network as target model

Figure 6: The KNN proxy can offer effective data debugging in various deep-learning scenarios.

compute the Shapley value. Especially since some interventions on the dataset might end up shifting the optimal path, and only by recomputing are we able to detect this shift.

**Deep learning pipelines.** We also measured the effectiveness of our approximation methods in several scenarios that involve deep learning models. In Figure 6a we use a pre-trained ResNet-18 model as the target model. We fine-tune it for 5 epochs on a noisy label dataset and see that Datascope KNN fares favorably compared to random label repair. Figure 6b shows the result of applying a pre-trained embedding model and evaluating both the Datascope KNN and the Datascope TMC approximations, where the KNN proxy again shows good performance. Finally, in Figure 6c we show how our method can be used to repair labels of a dataset used as a *support set* for a one-shot learning neural network. We use the matching networks model (Vinyals et al., 2016) which employs a learned "distance metric" between examples in the test set and those in the support set. This allows us to replace the standard Euclidean distance metric in our KNN proxy model with a custom one and achieve effective label repairs with efficiently computed Shapley values.

**Scalability.** We evaluate the speed of our algorithm for larger training datasets. We test the runtime for various sizes of the training set ($10k$-$1M$), the validation set ($100$-$10k$), and the number of features ($100$-$1k$). As expected, the impact of the training set size and validation set size is roughly linear (Figure 7). Even for large datasets, our method can compute Shapley scores in minutes.

Figure 7: Scalability analysis of our Datascope KNN Shapley algorithm over different training set, validation set, and feature vector sizes.

# 6 RELATED WORK

Targeted data repairs have been studied for some time now. Apart from the work mentioned in section 1, a notable piece of work is CleanLab which leverages confident learning to make targeted repairs of noisy labels (Northcutt et al., 2021). Our work focuses on the Shapley value given it was shown to be applicable to many scenarios (Jia et al., 2021). Apart from the data valuation scenario, the Shapley value has also been used for computing feature importance (Lundberg & Lee, 2017). On the other hand, the scope of our work is data importance over ML pipelines.

Debugging data pipelines has started receiving some attention recently. Systems such as Data X-Ray can debug data processing pipelines by finding groups of data errors that might have the same cause (Wang et al., 2015). Another example is `mlinspect` which also uses data provenance as an abstraction for automatically analyzing data preprocessing pipelines and discovering data distribution errors (Grafberger et al., 2022). A system called Rain leverages influence functions as a method for analyzing pipelines comprising of a model and a post-processing query (Wu et al., 2020). Rain also uses data provenance as a key ingredient, but their focus is on queries that take as input predictions of a model that has been trained directly on the source data.

# 7 CONCLUSION AND OUTLOOK

In this paper, we propose ease.ml/datascope, a framework for representing a wide range of ML pipelines that appear in real-world scenarios with the end goal of efficiently computing the Shapley value of source data examples. We show how this framework can be leveraged to provide significant speed-ups to Monte Carlo-based methods for Shapley value computation. Furthermore, we provide PTIME algorithms for computing the Shapley value using the KNN proxy model for several classes of ML pipelines. Finally, we empirically demonstrate that our methods achieve significant speed-ups over previously developed baselines while demonstrating competitive performance in a downstream data debugging task. Our code is available at github.com/easeml/datascope.

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

## A   Discussion about the Limitations of Prior Work

In this section, we provide a brief outline of the existing KNN approximation method for computing the Shapley value (Jia et al., 2019a) which was instrumental in laying the foundation for applying the KNN proxy model to Shapley computation. However, as we argue in this paper, this work is not directly applicable to ML pipelines as defined in this paper. Note that our goal here is to offer only intuition as to why it is the case, and thus we are leaving out many technical details. In Appendix F we present how the results in (Jia et al., 2019a) can be seen as a special case for computing Shapley values using the 1-NN proxy model.

The polynomial time approximation to computing Shapley values using the KNN proxy model established by Jia et al. (2019a) relies on several assumptions that do not hold in the context of fork/join pipelines. The prediction of the KNN model (and by extension its accuracy) for any training data (sub)set is strictly dependent on the labels of the top-$K$ data examples that are most similar to some validation example $t_{val}$ for which the KNN model is supposed to predict the label (and by extension result in a measurement of the accuracy of this prediction). To compute the Shapley value of a training data example $t_i \in \mathcal{D}_{tr}$, we need to know the accuracy difference (i.e. the marginal contribution) that occurs when adding that data example to every possible subset $\mathcal{D} \subseteq \mathcal{D}_{tr}$. In simple terms, the method in (Jia et al., 2019a) computes the Shapley value of an input data example by first sorting all data examples according to their similarity with $t_{val}$. After that, it relies on the observation that in order for $t_i$ to end up in the top-$K$ (and thus have a chance of impacting the prediction accuracy of some subset $\mathcal{D} \subseteq \mathcal{D}_{tr}$), of all data examples that are higher than $t_i$ in the sorting order, at most $(K-1)$ can be present in $\mathcal{D}$. It then computes how many subsets $\mathcal{D} \subseteq \mathcal{D}_{tr}$ of size $\alpha$ satisfy this condition. Specifically, if $t_i$ takes the $j$-th position in the sorting order, then the number of such subsets is $\binom{j-1}{K-1}\binom{|\mathcal{D}_{tr}|-j}{\alpha-K}$. Finally, it includes the Shapley weighing factor along with some combinatorial tricks to combine all this into a simple formula:

$$\varphi(t_i, t_{val}) = \sum_{j=1}^{|\mathcal{D}_{tr}|} \Big( m_T(y(t_i), t_{val}) - m_T(y(t_j), t_{val}) \Big) \binom{|\mathcal{D}_{tr}| - j}{j + 1}$$

As we can see, this method strictly expects that adding $t_i$ to any subset of $\mathcal{D}_{tr}$ will always result in either 0 or 1 data examples being added to the top-$K$ and that the choice between 0 and 1 strictly depends on the number of data examples that come before $t_i$ in the sorting order. Two core assumptions lie behind this expectation: (1) adding $t_i$ to a subset of $\mathcal{D}_{tr}$ will always result in exactly one additional data example being passed to KNN, and (2) the presence of any data example in the KNN training set is caused by the presence of exactly one data example in $\mathcal{D}_{tr}$. The first assumption allows us to separate data examples into those that come before $t_i$ in the sorting order and those that come after. The second assumption allows us to count subsets using binomial coefficients. If any of the two assumptions do not hold, then the simple combinatorial formula is no longer applicable because the data examples passed to KNN are no longer independent from each other. Map pipelines do not break these assumptions. On the other hand, fork pipelines break the first assumption, and join pipelines break both the first and the second assumption.

In this work, we examine the broader setting of ML pipelines which comes with several open questions. If any single training data examples $t_i \in \mathcal{D}_{tr}$ is associated with e.g. 10 data examples that are passed to KNN, and they are all intertwined in the sorting order, how do we efficiently compute the number of subsets $\mathcal{D} \subseteq \mathcal{D}_{tr}$ where adding a specific data example $t_i$ will result in altering the accuracy of the KNN prediction? If a data example that gets passed to KNN is the result of joining two data examples $t_{1,1}$ and $t_{2,1}$ from separate source datasets $\mathcal{D}_1$ and $\mathcal{D}_2$, but $t_{1,1}$ is also joined with other examples from $\mathcal{D}_2$ that make up even more output data examples, so removing $t_{2,1}$ from the training dataset will result in one data example not being passed to KNN but removing $t_{1,1}$ will result in more than one not being passed, how do we efficiently compute the number of subsets where adding $t_{1,1}$ will alter the KNN prediction? Do things change in the case of multi-class classification? Can we use model quality metrics other than accuracy? To answer these open questions, we employed all the theoretical components described in this paper, including provenance polynomials, ADD's, and model counting oracles. The theoretical insight we would like to convey is that all these components are fundamental to solving this problem and that this is the correct level of abstraction for analyzing ML pipelines and developing PTIME algorithms.

## B  DISCUSSION ABOUT TYPES OF ML PIPELINE OPERATORS

Here we provide an overview of types of pipeline operators that can be found in ML workflows. We base this discussion on operatos that can be found in the `scikit-learn` and `ML.NET` frameworks, as well as commonly used operators that can be found in real-world ML code.

**Unary Map:** These are functions that map single value inputs to single value outputs. Examples include:

- **Log** - Computes a logarithm of the input.
- **Missing Value Indicator** - returns a Boolean that indicates if the input is a missing value or not (e.g. `MissingIndicator` in `scikit-learn`).
- **Stopword Remover** - takes an input list of string tokens and removes the ones that correspond to stop-words (e.g. "the", "and", etc); the list of stop words is specified as an additional argument (e.g. `StopWordsRemovingTransformer` in `ML.NET`)

**Binary Numerical and Logical Map:** These are common mathematical operators such as addition, subtraction, multiplication, division, logical and, logical or, equality test, etc.

**Multi-Value Map:** Values containing multiple elements are taken as inputs and produced as outputs. A key example is a vector normalizing operator which maps a vector input to a vector output.

**Tuple Filter Map:** These operators remove tuples from the dataset based on the result of some unary map operation. Since these operators map a single tuple to either a single output tuple or to nothing, they are categorized as map filters. Examples include:

- **Missing Value Filter** - Removes tuples that contain missing values.
- **Range Filter** - Removes tuples where values of a specified column are outside a given range.

**Numerical Aggregate Reduce:** This operator takes an entire column and reduces it into a single numerical value. Examples include summation, counting, mean value, standard deviation, as well as minimal and maximal element selector operators.

**Unary Map with Reduce Elements:** These operator function similarly to regular *unary map* operators. However, their mapping operation is dependent on performing some *numerical aggregate reduce* operation beforehand. Examples include:

- **Min-Max Scaler** - Scales column values to a 0-1 range based on minimal and maximal element values which represent the pre-computed reduce element (e.g. `MinMaxScaler` in `scikit-learn`).
- **Standardization Scaler** - Same as the min-max scaler but transforms elements based on the pre-computed mean and standard deviation values (e.g. `StandardScaler` in `scikit-learn`).
- **One-Hot Encoder** - Encodes numerical features as a one-hot numerical array. Depends on a pre-computed list of unique column element values.
- **TD-IDF Encoder** - Converts textual values into their Term Frequency - Inverse Document Frequency encodings. This operator depends on a pre-computed dictionary of token frequencies.

**Data Augmentation Fork:** This can be any data augmentation operator that maps input tuples to some specified number of output tuples. Examples include: random noise injection, randomly shifting or rotating images, removing or replacing characters in text to simulate misspelling, etc.

**One-to-Many Join:** Join operators compute a matching between two sets of tuples $\mathcal{D}_A$ and $\mathcal{D}_B$, and for each pair of matched input tuples produce a single output tuple. In general there are no constraints on the kinds of matchings that can be performed. However, the specific type of join we describe here, referred to as one-to-many type join requires that tuples from one of the two sets (e.g. $\mathcal{D}_A$) can be matched with at most one tuple from the other set (e.g. $\mathcal{D}_B$). At the same time, tuples from $\mathcal{D}_B$ can be matched with multuple tuples from $\mathcal{D}_A$.

## C  PRELIMINARY: ADDITIVE DECISION DIAGRAMS (ADD'S)

In this section, we describe a type of decision diagram that we use as a tool for compact representation of functions over Boolean inputs. The process of translating functions into data structures for easier analysis is referred to as *knowledge compilation*. We briefly describe this in the context of our work, and then go over the data structure we use in our methods – Additive Decision Diagrams.

**Knowledge Compilation.** Our approach to computing the Shapley value will rely upon being able to construct functions over Boolean inputs $\phi : \mathcal{V}_A \to \mathcal{E}$, where $\mathcal{E}$ is some finite *value set*. We require an elementary algebra with $+$, $-$, $\cdot$ and $/$ operations to be defined for this value set. Furthermore, we require this value set to contain a *zero element* $0$, as well as an *invalid element* $\infty$ representing

an undefined result (e.g. a result that is out of bounds). We then need to count the number of value assignments $v \in \mathcal{V}_A$ such that $\phi(v) = e$, for some specific value $e \in \mathcal{E}$. This is referred to as the *model counting* problem, which is #P complete for arbitrary logical formulas Valiant (1979); Arora & Barak (2009). For example, if $A = \{a_1, a_2, a_3\}$, we can define $\mathcal{E} = \{0, 1, 2, 3, \infty\}$ to be a value set and a function $\phi(v) := v(a_1) + v(a_2) + v(a_3)$ corresponding to the number of variables in $A$ that are set to 1 under some value assignment $v \in \mathcal{V}_A$.

*Knowledge compilation* Cadoli & Donini (1997) has been developed as a well-known approach to tackle this model counting problem. It was also successfully applied to various problems in data management Jha & Suciu (2011). One key result from this line of work is that, if we can construct certain polynomial-size data structures to represent our logical formula, then we can perform model counting in polynomial time. Among the most notable of such data structures are *decision diagrams*, specifically binary decision diagrams Lee (1959); Bryant (1986) and their various derivatives Bahar et al. (1997); Sanner & McAllester (2005); Lai et al. (1996). For our purpose in this paper, we use the *additive decision diagrams* (ADD), as detailed below.

**Additive Decision Diagrams (ADD).** We define a simplified version of the *affine algebraic decision diagrams* Sanner & McAllester (2005). An ADD is a directed acyclic graph defined over a set of nodes $\mathcal{N}$ and a special *sink node* denoted as $\boxdot$. Each node $n \in \mathcal{N}$ is associated with a variable $a(n) \in A$. Each node has two outgoing edges, $c_L(n)$ and $c_H(n)$, that point to its *low* and *high* child nodes, respectively. For some value assignment $v$, the low/high edge corresponds to $v(a) = 0/v(a) = 1$. Furthermore, each low/high edge is associated with an increment $w_L/w_H$ that maps edges to elements of $\mathcal{E}$.

Note that each node $n \in \mathcal{N}$ represents the root of a subgraph and defines a Boolean function. Given some value assignment $v \in \mathcal{V}_A$ we can evaluate this function by constructing a path starting from $n$ and at each step moving towards the low or high child depending on whether the corresponding variable is assigned 0 or 1. The value of the function is the result of adding all the edge increments together. Figure 8a presents an example ADD with one path highlighted in red. Formally, we can define the evaluation of the function defined by the node $n$ as follows:

$$\mathrm{eval}_v(n) := \begin{cases} 0, & \text{if } n = \boxdot, \\ w_L(n) + \mathrm{eval}_v(c_L(n)) & \text{if } v(x(n)) = 0, \\ w_H(n) + \mathrm{eval}_v(c_H(n)) & \text{if } v(x(n)) = 1. \end{cases} \tag{9}$$

In our work, we focus specifically on ADD's that are *full* and *ordered*. A diagram is full if every path from root to sink encounters every variable in $A$ exactly once. For example, in Figure 8a we see a full diagram over the set of variables $A = \{a_{1,1}, a_{1,2}, a_{2,1}, a_{2,2}, a_{2,3}\}$. If any of the variables in $A$ has no node associated with it, then the diagram is not considered full. On the other hand, an ADD is ordered when on each path from root to sink variables always appear in the same order. For this purpose, we define $\pi : A \to \{1, ..., |A|\}$ to be a permutation of variables that assigns each variable $a \in A$ an index. For example, in Figure 8a, the variable order is $\pi = \{a_{1,1} \mapsto 1, a_{1,2} \mapsto 4, a_{2,1} \mapsto 2, a_{2,2} \mapsto 3, a_{2,3} \mapsto 5\}$. It is possible, for example, to swap the two nodes on the left side that correspond to $a_{2,1}$ and $a_{2,2}$. This, however, makes the diagram unordered, which dramatically complicates certain operations (e.g. the diagram summation operation that we will describe shortly).

**Diagram Diameter.** We define the diameter of an ADD as the maximum number of nodes associated with any single variable. Formally we can write:

$$\mathrm{diam}(\mathcal{N}) := \max_{a_i \in A} \left| \{n \in \mathcal{N} : a(n) = a_i\} \right| \tag{10}$$

We can immediately notice that the size of any ADD with a set of nodes $\mathcal{N}$ and variables $A$ is bounded by $O(|A| \cdot \mathrm{diam}(\mathcal{N}))$.

**Model Counting.** We define a model counting operator

$$\mathrm{count}_e(n) := \left| \left\{ v \in \mathcal{V}_{A[\leq \pi(a(n))]} \mid \mathrm{eval}_v(n) = e \right\} \right|, \tag{11}$$

where $A[\leq \pi(a(n))]$ is the subset of variables in $A$ that include $a(n)$ and all variables that come before it in the permutation $\pi$. For an ordered and full ADD, $\mathrm{count}_e(n)$ satisfies the following recursion:

$$\mathrm{count}_e(n) := \begin{cases} 1, & \text{if } e = 0 \text{ and } n = \boxdot, \\ 0, & \text{if } e = \infty \text{ or } n = \boxdot, \\ \mathrm{count}_{e-w_L(n)}(c_L(n)) + \mathrm{count}_{e-w_H(n)}(c_H(n)), & \text{otherwise.} \end{cases} \tag{12}$$

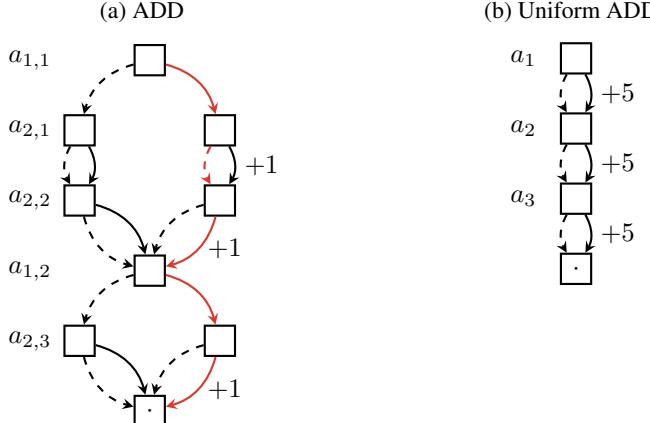

Figure 8: (a) An ordered and full ADD for computing $\phi(v) := v(a_{1,1}) \cdot \big(v(a_{2,1}) + v(a_{2,2})\big) + v(a_{1,2}) \cdot v(a_{2,3})$. (b) A uniform ADD for computing $\phi(v) := 5 \cdot (v(a_1) + v(a_2) + v(a_3))$.

The above recursion can be implemented as a dynamic program with computational complexity $O(|\mathcal{N}| \cdot |\mathcal{E}|)$.

Figure 8b shows a special case of a full and ordered ADD, which we call a *uniform* ADD. It is structured as a chain with one node per variable, where all low increments equal zero and all high increments equal some constant $E \in \mathcal{E}$. For this type of ADD, we can perform model counting in constant time, assuming that we have a precomputed table of factorials of size $O(|\mathcal{N}|)$ that allows us to compute binomial coefficients in constant time. The $\mathrm{count}_e$ operator for a uniform ADD can be defined as

$$\mathrm{count}_e(n) := \begin{cases} \binom{\pi(a(n))}{e/E}, & \text{if } e \bmod E = 0, \\ 0 & \text{otherwise.} \end{cases} \tag{13}$$

Intuitively, if we observe the uniform ADD shown in Figure 8b, we see that the result of an evaluation must be a multiple of $5$. For example, to evaluate to $10$, the evaluation path must pass a *high* edge exactly twice. Therefore, in a 3-node ADD with root node $n_R$, the result of $\mathrm{count}_{10}(n_R)$ will be exactly $\binom{3}{2}$.

**Special Operations on ADD's.** Given an ADD with node set $\mathcal{N}$, we define two operations that will become useful later on when constructing diagrams for our specific scenario:

1. *Variable restriction*, denoted as $\mathcal{N}[a_i \leftarrow V]$, which restricts the domain of variables $A$ by forcing the variable $a_i$ to be assigned the value $V$. This operation removes every node $n \in \mathcal{N}$ where $a(n) = a_i$ and rewires all incoming edges to point to the node's high or low child, depending on whether $V = 1$ or $V = 0$. The resulting diagram will have between $1$ and $\mathrm{diam}(\mathcal{N})$ nodes less than the original diagram, depending on the number of nodes associated with variable $a_i$.

2. *Diagram summation*, denoted as $\mathcal{N}_1 + \mathcal{N}_2$, where $\mathcal{N}_1$ and $\mathcal{N}_2$ are two ADD's over the same (ordered) set of variables $A$. ordered in the same way. It starts from the respective root nodes $n_1$ and $n_2$, and produces a new node $n := n_1 + n_2$. We then apply the same operation to child nodes. Therefore, $c_L(n_1 + n_2) := c_L(n_1) + c_L(n_2)$ and $c_H(n_1 + n_2) := c_H(n_1) + c_H(n_2)$. Also, for the increments, we can define $w_L(n_1 + n_2) := w_L(n_1) + w_L(n_2)$ and $w_H(n_1 + n_2) := w_H(n_1) + w_H(n_2)$. The size of the resulting diagram is bounded by $O(|A| \cdot \mathrm{diam}(\mathcal{N}_1) \cdot \mathrm{diam}(\mathcal{N}_2))$. A proof of this claim is presented in subsection E.2.

## D    CONSTRUCTING POLYNOMIAL-SIZE ADD'S FOR ML PIPELINES

Algorithm 1 presents our main procedure COMPILEADD that constructs an ADD for a given dataset $\mathcal{D}$ made up of tuples annotated with provenance polynomials. This is achieved by invoking the procedure COMPILEADD($\mathcal{D}, A, t_i, t_{val}$) constructs an ADD with node set $\mathcal{N}'$ that computes

$$\phi(v \mid t_i, t_{val}, t') := \begin{cases} \infty, & \text{if } t' \notin \mathcal{D}[v]), \\ \mathrm{tally}(\mathcal{D}[v] \mid t', t_{val}), & \text{otherwise.} \end{cases} \tag{14}$$

---

**Algorithm 1** Compiling a provenance-tracked dataset into ADD.

1: **function** COMPILEADD
2:     **inputs**
3:         $\mathcal{D}$, provenance-tracked dataset;
4:         $A$, set of variables;
5:         $t_i$, boundary tuple;
6:         $t_{val}$, validation tuple;
7:     **outputs**
8:         $\mathcal{N}$, nodes of the compiled ADD;
9: **begin**
10:     $\mathcal{N} \leftarrow \{\}$
11:     $\mathcal{P} \leftarrow \{(a_1, a_2) \in A \ : \exists t_i \in \mathcal{D}, a_1 \in p(t_i) \ \wedge \ a_2 \in p(t_i)\}$
12:     $A_L \leftarrow$ GETLEAFVARIABLES($\mathcal{P}$)
13:     **for** $A_C \in$ GETCONNECTEDCOMPONENTS($\mathcal{P}$) **do**
14:         $A' \leftarrow A_C \setminus A_L$
15:         $\mathcal{N}' \leftarrow$ CONSTRUCTADDTREE($A'$)
16:         $\mathcal{D}' \leftarrow \{t' \in \mathcal{D} \ : \ p(t') \cup A_C \neq \emptyset\}$
17:         **for** $v \in \mathcal{V}_{A'}$ **do**
18:             $\mathcal{N}_C \leftarrow$ CONSTRUCTADDCHAIN($A_C \cap A_L$)
19:             **for** $n \in \mathcal{N}_C$ **do**
20:                 $v' \leftarrow v \cup \{a(n) \rightarrow 1\}$
21:                 $w_H(n) \leftarrow |\{t' \in \mathcal{D}' \ : \ \text{eval}_{v'} p(t') = 1 \ \wedge \ \sigma(t', t_{val}) \geq \sigma(t_i, t_{val})\}|$
22:             **end for**
23:             $\mathcal{N}' \leftarrow$ APPENDTOADDPATH($\mathcal{N}', \mathcal{N}_C, v$)
24:         **end for**
25:         $\mathcal{N} \leftarrow$ APPENDTOADDROOT($\mathcal{N}, \mathcal{N}'$)
26:     **end for**
27:     **for** $a' \in p(t)$ **do**
28:         **for** $n \in \mathcal{N}$ **where** $a(n) = a'$ **do**
29:             $w_L(n) \leftarrow \infty$
30:         **end for**
31:     **end for**
32:     **return** $\mathcal{N}$
33: **end function**

---

We provide a more detailed description of Algorithm 1 in subsection D.1.

To construct the function defined in Equation 8, we need to invoke COMPILEADD once more by passing $t''$ instead of $t'$ in order to obtain another diagram $\mathcal{N}''$. The final diagram is obtained as a result of $\mathcal{N}'[a_i \leftarrow 0] + \mathcal{N}''[a_i \leftarrow 1]$. In other words, by performing a *diagram summation* operation over diagrams $\mathcal{N}'$ (with *variable restriction* $a_i \leftarrow 0$) and $\mathcal{N}''$ (with *variable restriction* $a_i \leftarrow 1$). The size of the resulting diagram will still be bounded by $O(|\mathcal{D}|)$.

We can now examine different types of canonical pipelines and see how their structures are reflected onto the structure of ADD's. In summary, we can construct an ADD with polynomial-size for canonical pipelines, and therefore, by Theorem 4.2, the computation of the corresponding counting oracles is in PTIME.

**One-to-Many Join Pipeline.** In a *star* database schema, this corresponds to a *join* between a *fact* table and a *dimension* table, where each tuple from the dimension table can be joined with multiple tuples from the fact table. It can be represented by an ADD similar to the one in Figure 8a.

**Corollary D.1.** *For the $K$-NN accuracy utility and a one-to-many* join *pipeline, which takes as input two datasets, $\mathcal{D}_F$ and $\mathcal{D}_D$, of total size $|\mathcal{D}_F| + |\mathcal{D}_D| = N$ and outputs a joined dataset of size $O(N)$, the Shapley value can be computed in $O(N^4)$ time.*

*Proof.* This follows from the observation that in Algorithm 1, each connected component $A_C$ will be made up of one variable corresponding to the dimension table and one or more variables corresponding to the fact table. Since the fact table variables will be categorized as "leaf variables", the expression $A_C \setminus A_L$ in Line 14 will contain only a single element – the dimension table variable. Consequently, the ADD tree in $\mathcal{N}'$ will contain a single node. On the other side, the $A_C \cap A_L$ expression will

contain all fact table variables associated with that single dimension table variable. That chain will be added to the ADD tree two times for two outgoing branches of the single tree node. Hence, the ADD segment will be made up of two fact table variable chains stemming from a single dimension table variable node. There will be $O(|\mathcal{D}_D|)$ partitions in total. Given that the fact table variables are partitioned, the cumulative size of their chains will be $O(|\mathcal{D}_F|)$. Therefore, the total size of the ADD with all partitions joined together is bounded by $O(|\mathcal{D}_D| + |\mathcal{D}_F|) = O(N)$.

Given fact and combining it with Theorem 4.2 we know that the counting oracle can be computed in time $O(N)$ time. Finally, given Theorem 4.1 and the structure of Equation 6 we can observe that the counting oracle is invoked $O(N^3)$ times. As a result, we can conclude that the total complexity of computing the Shapley value is $O(N^4)$. Here, we assume that we have a precomputed table of factorials from 1 to $N$ that allows us to compute the binomial coefficient in constant time. $\qquad\square$

**Fork Pipeline.** The key characteristic of a pipeline $f$ that contains only *fork* or *map* operators is that the resulting dataset $f(\mathcal{D})$ has provenance polynomials with only a single variable. This is due to the absence of joins, which are the only operator that results in provenance polynomials with a combination of variables.

**Corollary D.2.** *For the $K$-NN accuracy utility and a* fork *pipeline, which takes as input a dataset of size $N$ and outputs a dataset of size $M$, the Shapley value can be computed in $O(M^2N^2)$ time.*

*Proof.* The key observation here is that, since all provenance polynomials contain only a single variable, there is no interdependency between them, which means that the connected components returned in Line 13 of Algorithm 1 will each contain a single variable. Therefore, the size of the resulting ADD will be $O(N)$. Consequently, similar to the proof of the previous corollary, the counting oracle can be computed in time $O(N)$ time. In this case, the size of the output dataset is $O(M)$ which means that Equation 6 will invoke the oracle $O(M^2N)$ times. Therefore, the total time complexity of computing the Shapley value will be $O(M^2N^2)$. Here, we assume that we have a precomputed table of factorials from 1 to $N$ that allows us to compute the binomial coefficient in constant time. $\qquad\square$

**Map Pipeline.** A *map* pipeline is similar to *fork* pipeline in the sense that every provenance polynomial contains only a single variable. However, each variable now can appear in a provenance polynomial of *at most* one tuple, in contrast to *fork* pipeline where a single variable can be associated with *multiple* tuples. This additional restriction results in the following corollary:

**Corollary D.3.** *For the $K$-NN accuracy utility and a* map *pipeline, which takes as input a dataset of size $N$, the Shapley value can be computed in $O(N^2)$ time.*

*Proof.* There are two arguments we need to make which will result in the reduction of complexity compared to fork pipelines. The first argument is that given that each variable can appear in the provenance polynomial of at most one tuple, having its value set to 1 can result in either zero or one tuple contributing to the top-$K$ tally. It will be one if that tuple is more similar than the boundary tuple $t$ and it will be zero if it is less similar. Consequently, our ADD will have a chain structure with high-child increments being either 0 or 1. If we partition the ADD into two chains, one with all increments 1 and another with all increments 0, then we end up with two uniform ADD's. As shown in Equation 13, model counting of uniform ADD's can be achieved in constant time. The only difference here is that, since we have to account for the support size of each model, computing the oracle $\omega(\alpha, \gamma', \gamma'' | t_i, t_{val}, t', t'')$ for a given $\alpha$ will require us to account for different possible ways to split $\alpha$ across the two ADD's. However, since the tuple $t$ needs to be the boundary tuple, which means it is the $K$-th most similar, there need to be exactly $K - 1$ variables from the ADD with increments 1 that can be set to 1. This gives us a single possible distribution of $\alpha$ across two ADD's. Hence, the oracle can be computed in constant time.

As for the second argument, we need to make a simple observation. For map pipelines, given a boundary tuple $t'$ and a tally vector $\gamma'$ corresponding to the variable $a_i$ being assigned the value 0, we know that setting this variable to 1 can introduce at most one tuple to the top-$K$. That could only be the single tuple associated with $a_i$. If this tuple has a lower similarity score than $t'$, there will be no change in the top-$K$. On the other side, if it has a higher similarity, then it will become part of the top-$K$ and it will evict exactly $t'$ from it. Hence, there is a unique tally vector $\gamma''$ resulting from $a_i$ being assigned the value 1. This means that instead of computing the counting oracle $\omega(\alpha, \gamma', \gamma'' | t_i, t_{val}, t', t'')$, we can compute the oracle $\omega(\alpha, \gamma' | t_i, t_{val}, t')$. This means that, in Equation 6 we can eliminate the iteration over $t''$ which saves us an order of $O(N)$ in complexity.

As a result, Equation 6 will make $O(N^2)$ invocations to the oracle which can be computed in constant time. Here, we assume that we have a precomputed table of factorials from 1 to $N$ that allows us to compute the binomial coefficient in constant time. Hence, the final complexity of computing the Shapley value will be $O(N^2)$. □

### D.1 Details of Algorithm 1

In this section, we examine the method of compiling a provenance-tracked dataset $f(\mathcal{D}_{tr})$ that results from a pipeline $f$. The crux of the method is defined in Algorithm 1 which is an algorithm that takes a dataset $f(\mathcal{D}_{tr})$ with provenance tracked over a set of variables $A$, a boundary tuple $t' \in f(\mathcal{D}_{tr})$ and a validation tuple $t_{val} \in f(\mathcal{D}_{val})$. The result is an ADD that computes the following function:

$$\phi(v \mid t_i, t_{val}, t') := \begin{cases} \infty, & \text{if } t' \notin f(\mathcal{D}_{tr}[v]), \\ \text{tally}(f(\mathcal{D}_{tr}[v]) \mid t', t_{val}), & \text{otherwise.} \end{cases} \tag{15}$$

Assuming that all provenance polynomials are actually a single conjunction of variables and that the tally is always a sum over those polynomials, it tries to perform factoring by determining if there are any variables that can be isolated. This is achieved by first constructing the set of "leaf variables" $A_L$ (Line 12). No pair of variables in $A_L$ ever appears in the same provenance polynomial. In graph theory, this set is also known as the "independent set". We use a heuristic approach to construct this set that prioritizes the least frequently occurring variables and completes them in $O(N)$ time. We then iterate over each "connected component" $A_C$ (Line 13) where any two variables are "connected" if they are ever in the same provenance polynomial. Then we get the set $A' = A_C \setminus A_L$ which contains variables that cannot be isolated (because they appear in polynomials in multiple tuples with multiple different variables). We form a group that will be treated as one binary vector and based on the value of that vector we would take a specific path in the tree. We thus take the group of variables and call the CONSTRUCTADDTREE function to construct an ADD tree (Line 15).

Every path in this tree corresponds to one value assignment to the variables in that tree. Then, for every path we call the CONSTRUCTADDCHAIN to build a chain made up of the isolated variables and call APPENDTOADDPATH to append them to the leaf of that path (Line 23). For each variable in the chain, we also define an increment that is defined by the number of tuples that will be more similar than the boundary tuple $t'$ and also have their provenance polynomial "supported" by the path. We thus construct a segment of the final ADD made up of different components. We append this segment to the final ADD using the APPENDTOADDROOT function. We don't explicitly define these functions but we illustrate their functionality in Figure 9.

## E Additional Proofs and Details

### E.1 Proof of Theorem 4.1

*Proof.* This theorem can easily be proven by observing the structure of the Shapley value of a tuple $t_i$ for a single validation tuple $t_{val}$, as we defined it in Equation 6:

$$\varphi(t_i, t_{val}) = \frac{1}{|A|} \sum_{t', t'' \in f(\mathcal{D}_{tr})} \sum_{\alpha=1}^{|A|} \binom{|A|-1}{\alpha}^{-1} \sum_{\gamma', \gamma'' \in \Gamma_{\mathcal{Y}, K}} m_\Delta(\gamma', \gamma'' \mid t_{val}) \cdot \omega(\alpha, \gamma', \gamma'' \mid t_i, t_{val}, t', t'').$$

We can notice that it is made up of several sums: (1) the left-most one is a sum over $t', t'' \in f(\mathcal{D}_{tr})$ which for a canonical pipeline $f$ is a set of cardinality in $O(|\mathcal{D}_{tr}|)$; (2) the next one is a sum over $|A|$ elements which is $O(|\mathcal{D}_{tr}|)$ according to the definition of $A$ given in subsection 3.1; and finally (3) the right-most sum is over $\gamma', \gamma'' \in \Gamma_{\mathcal{Y}, K}$ where $\Gamma_{\mathcal{Y}, K}$ is the set of all $|\mathcal{Y}|$-dimensional label tally vectors which can be defined as $\Gamma_{\mathcal{Y}, K} := \{\gamma \in \mathbb{N}^{|\mathcal{Y}|} : K \geq \sum_i \gamma_i\}$ and can be treated as constant since it does not depend on $|\mathcal{D}_{tr}|$. As we can see, given that all sums in $\varphi(t_i, t_{val})$ are $O(|\mathcal{D}_{tr}|)$, then it is safe to conclude that if we can compute $\omega(\alpha, \gamma', \gamma'' \mid t_i, t_{val}, t', t'')$ in time polynomial w.r.t $|\mathcal{D}_{tr}|$, then we can also compute $\varphi(t_i, t_{val})$ in time polynomial in $|\mathcal{D}_{tr}|$. Finally, as mentioned in , the Shapley value for a tuple $t_i$ can be computed as $\varphi(t_i) = w \cdot \sum_{t_{val} \in f(\mathcal{D}_{val})} \varphi(t_i, t_{val})$ which contains a sum over $O(|\mathcal{D}_{val}|)$ elements (given that the pipeline $f$ is canonical). Hence, we can see that the Shapley value can be computed in time polynomial in $|\mathcal{D}_{tr}|$ and $|\mathcal{D}_{val}|$, which concludes our proof. □

### E.2 Proof of Theorem 4.2

**Model Counting for ADD's.** We start off by proving that Equation 12 correctly performs model counting.

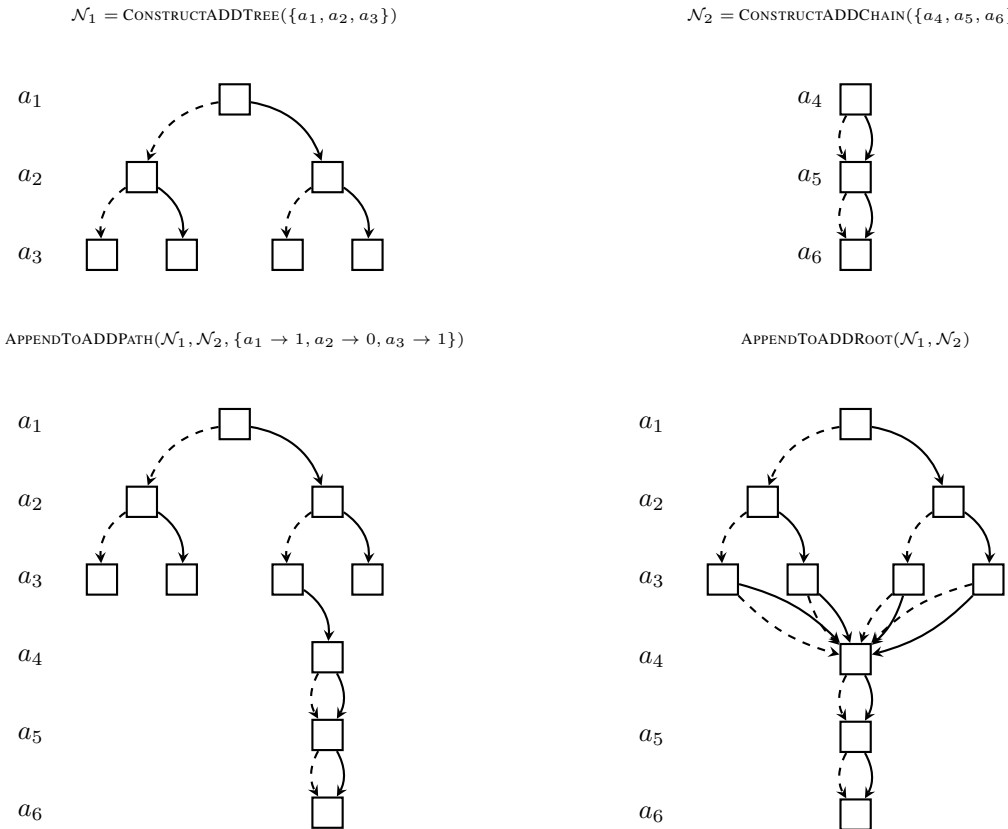

Figure 9: An example of ADD compilation functions.

**Lemma E.1.** *For a given node $n \in \mathcal{N}$ of an ADD and a given value $e \in \mathcal{E}$, Equation 12 correctly computes* $\text{count}_e(n)$ *which returns the number of assignments* $v \in \mathcal{V}_A$ *such that* $\text{eval}_v(n) = e$. *Furthermore, when computing* $\text{count}_e(n)$ *for any* $n \in \mathcal{N}$, *the number of computational steps is bounded by* $O(|\mathcal{N}| \cdot |\mathcal{E}|)$.

*Proof.* We will prove this by induction on the structure of the recursion.

*(Base case.)* Based on Equation 9, when $n = \square$ we get $\text{eval}_v(n) = 0$ for all $v$. Furthermore, when $n = \square$, the set $\mathcal{V}_A[a_{>\pi(a(n))} = 0]$ contains only one value assignment with all variables set to zero. Hence, the model count will equal to 1 only for $e = 0$ and it will be 0 otherwise, which is reflected in the base cases of Equation 12.

*(Inductive step.)* Because our ADD is ordered and full, both $c_L(n)$ and $c_H(n)$ are associated with the same variable, which is the predecessor of $a(n)$ in the permutation $\pi$. Based on this and the induction hypothesis, we can assume that

$$
\begin{aligned}
\text{count}_{e-w_L(n)}(c_L(n)) &= \left| \left\{ v \in \mathcal{V}_{A[\leq a(c_L(n))]} \mid \text{eval}_v(c_L(n)) = e - w_L(n) \right\} \right| \\
\text{count}_{e-w_H(n)}(c_H(n)) &= \left| \left\{ v \in \mathcal{V}_{A[\leq a(c_H(n))]} \mid \text{eval}_v(c_H(n)) = e - w_H(n) \right\} \right|
\end{aligned}
\tag{16}
$$

We would like to compute $\text{count}_e(n)$ as defined in Equation 11. It computes the size of a set defined over possible value assignments to variables in $A[\leq a(n)]$. The set of value assignments can be partitioned into two distinct sets: one where $a(n) \leftarrow 0$ and one where $a(n) \leftarrow 1$. We thus obtain the following expression:

$$
\begin{aligned}
\text{count}_e(n) :=& \left| \left\{ v \in \mathcal{V}_{A[\leq a(n)]}\big[a(n) \leftarrow 0\big] \mid \text{eval}_v(n) = e \right\} \right| \\
&+ \left| \left\{ v \in \mathcal{V}_{A[\leq a(n)]}\big[a(n) \leftarrow 1\big] \mid \text{eval}_v(n) = e \right\} \right|
\end{aligned}
\tag{17}
$$

Based on Equation 9, we can transform the $\text{eval}_v(n)$ expressions as such:

$$\text{count}_e(n) := \left| \left\{ v \in \mathcal{V}_{A[\leq a(c_L(n))]} \mid w_L(n) + \text{eval}_v(c_L(n)) = e \right\} \right| \tag{18}$$
$$+ \left| \left\{ v \in \mathcal{V}_{A[\leq a(c_L(n))]} \mid w_H(n) + \text{eval}_v(c_H(n)) = e \right\} \right|$$

Finally, we can notice that the set size expressions are equivalent to those in Equation 16. Therefore, we can obtain the following expression:

$$\text{count}_e(n) := \text{count}_{e-w_L(n)}(c_L(n)) + \text{count}_{e-w_H(n)}(c_H(n)) \tag{19}$$

which is exactly the recursive step in Equation 12. This concludes our inductive proof and we move onto proving the complexity bound.

*(Complexity.)* This is trivially proven by observing that since $\text{count}$ has two arguments, we can maintain a table of results obtained for each $n \in \mathcal{N}$ and $e \in \mathcal{E}$. Therefore, we know that we will never need to perform more than $O(|\mathcal{N}| \cdot |\mathcal{E}|)$ invocations of $\text{count}_e(n)$.

$\square$

**ADD Construction.** Next, we prove that the size of an ADD resulting from *diagram summation* as defined in Appendix C is linear in the number of variables.

The size of the diagram resulting from a sum of two diagrams with node sets $\mathcal{N}_1$ and $\mathcal{N}_2$ can be loosely bounded by $O(|\mathcal{N}_1| \cdot |\mathcal{N}_2|)$ assuming that its nodes come from a combination of every possible pair of operand nodes. However, given the much more narrow assumptions we made in the definition of the node sum operator, we can make this bound considerably tighter. As mentioned in Appendix C, the size of any ADD with set of nodes $\mathcal{N}$ and variables $A$ is bounded by $O(|A| \cdot \text{diam}(\mathcal{N}))$. We can use this fact to prove a tighter bound on the size of an ADD resulting from a sum operation:

**Lemma E.2.** *Given two full ordered ADD's with nodes $\mathcal{N}_1$ and $\mathcal{N}_2$, noth defined over the set of variables $A$, the number of nodes in $\mathcal{N}_1 + \mathcal{N}_2$ is bounded by $O(|A| \cdot \text{diam}(\mathcal{N}_1) \cdot \text{diam}(\mathcal{N}_2))$.*

*Proof.* It is sufficient to show that $\text{diam}(\mathcal{N}_1 + \mathcal{N}_2) = O(\text{diam}(\mathcal{N}_1) \cdot \text{diam}(\mathcal{N}_2))$. This is a direct consequence of the fact that for full ordered ADD's the node sum operator is defined only for nodes associated with the same variable. Since the only way to produce new nodes is by merging one node in $\mathcal{N}_1$ with one node in $\mathcal{N}_2$, and given that we can merge nodes associated with the same variable, the number of nodes associated with the same variable in the resulting ADD equals the product of the corresponding number of nodes in the constituent ADD's. Since the diameter is simply the upper bound of the number of nodes associated with any single variable, the same upper bound in the resulting ADD cannot be larger than the product of the upper bounds of constituent nodes. $\square$

**Computing the Oracle using ADD's.** Finally, we prove the correctness of Theorem 4.2.

**Lemma E.3.** *Given an Additive Decision diagram with root node $n(t_i, t_{val}, t', t'')$ that computes the Boolean function $\phi(v|t_i, t_{val}, t', t'')$ as defined in Equation 8, the counting oracle $\omega(\alpha, \gamma', \gamma''|t_i, t_{val}, t', t'')$ defined in Equation 7 can be computed as:*

$$\omega(\alpha, \gamma', \gamma''|t_i, t_{val}, t', t'') := \text{count}_{(\alpha, \gamma', \gamma'')}(n(t_i, t_{val}, t', t'')) \tag{20}$$

*Proof.* Given a training dataset $\mathcal{D}_{tr}$ and a data preprocessing pipeline $f$, we have $f(\mathcal{D}_{tr})$ as the output of that pipeline and input to an ML model. Let us define $f(\mathcal{D}_{tr})[\geq_{\sigma(\cdot, t_{val})} t'] \subseteq f(\mathcal{D}_{tr})$ as a set of tuples in $f(\mathcal{D}_{tr})$ with similarity to a validation tuple $t_{val}$ higher or equal than that of $t'$, formally $f(\mathcal{D}_{tr})[\geq_{\sigma(\cdot, t_{val})} t'] := \{t'' \in f(\mathcal{D}_{tr}) : \sigma(t'', t_{val}) \geq \sigma(t', t_{val})\}$. Similarly to $f(\mathcal{D}_{tr})$, the semantics of $f(\mathcal{D}_{tr})[\geq_{\sigma(\cdot, t_{val})} t']$ is also that of a set of possible candidate sets. Given a value assignment $v$, we can obtain $f(\mathcal{D}_{tr}[v])[\geq_{\sigma(\cdot, t_{val})} t']$ from $f(\mathcal{D}_{tr}[v])$. For convenience, we also define $f(\mathcal{D}_{tr})[\geq_{\sigma(\cdot, t_{val})} t'][=_\ell y]$ as a subset of $f(\mathcal{D}_{tr})[\geq_{\sigma(\cdot, t_{val})} t']$ with only tuples that have label $y$. Given these definitions, we can define several equivalences. First, for $\text{top}_K$ we have:

$$\left( t' = \text{top}_K\big(f(\mathcal{D}_{tr}[v]) \mid t_{val}\big) \right) \iff \left( t' \in f(\mathcal{D}_{tr}[v]) \wedge \big|f(\mathcal{D}_{tr}[v])[\geq_{\sigma(\cdot, t_{val})} t']\big| = K \right) \tag{21}$$

In other words, for $t'$ to be the tuple with the $K$-th highest similarity in $f(\mathcal{D}_{tr}[v])$, it needs to be a member of $f(\mathcal{D}_{tr}[v])$ and the number of tuples with similarity greater or equal to $t'$ has to be exactly $K$. Similarly, we can define the equivalence for $\text{tally}(\cdot|t', t_{val})$:

$$\left( \gamma' = \text{tally}\big(f(\mathcal{D}_{tr}[v]) \mid t', t_{val}\big) \right) \iff \left( \forall y \in \mathcal{Y}, \gamma'_y = \big|f(\mathcal{D}_{tr}[v])[\geq_{\sigma(\cdot, t_{val})} t'][=_\ell y]\big| \right) \tag{22}$$

This is simply an expression that partitions the set $f(\mathcal{D}_{tr}[v])[\geq_{\sigma(\cdot, t_{val})} t']$ based on $y$ and tallies them up. The next step is to define an equivalence for $(t' = \text{top}_K(f(\mathcal{D}_{tr}[v]) \mid t_{val})) \wedge (\gamma' = \text{tally}(f(\mathcal{D}_{tr}[v]) \mid t', t_{val}))$. We can notice that since $|\gamma'| = K$, if we have $(\forall y \in \mathcal{Y}, \gamma'_y = |f(\mathcal{D}_{tr}[v])[\geq_{\sigma(\cdot, t_{val})} t'][=_\ell y]|)$ then we can conclude that $(|f(\mathcal{D}_{tr}[v])[\geq_{\sigma(\cdot, t_{val})} t']| = K)$ is redundant. Hence, we can obtain:

$$\left( t' = \text{top}_K\big(f(\mathcal{D}_{tr}[v]) \mid t_{val}\big) \right) \wedge \left( \gamma' = \text{tally}\big(f(\mathcal{D}_{tr}[v]) \mid t', t_{val}\big) \right) \iff \left( t' \in f(\mathcal{D}_{tr}[v]) \right) \wedge \left( \forall y \in \mathcal{Y}, \gamma_y = |f(\mathcal{D}_{tr}[v])[\geq_\sigma t][=_\ell y]| \right) \tag{23}$$

According to Equation 22, we can reformulate the right-hand side of the above equivalence as:

$$\left( t' = \text{top}_K\big(f(\mathcal{D}_{tr}[v]) \mid t_{val}\big) \right) \wedge \left( \gamma' = \text{tally}\big(f(\mathcal{D}_{tr}[v]) \mid t', t_{val}\big) \right) \iff \left( t' \in f(\mathcal{D}_{tr})[v] \right) \wedge \left( \gamma' = \text{tally}\big(f(\mathcal{D}_{tr}[v]) \mid t', t_{val}\big) \right) \tag{24}$$

We can construct a similar expression for $t'$ and $v[a_i = 1]$ so we cover four out of five predicates in Equation 7. The remaining one is simply the support of the value assignment $v$ which we will leave intact. This leaves us with the following equation for the counting oracle:

$$\omega(\alpha, \gamma', \gamma'' \mid t_i, t_{val}, t', t'') := \sum_{v \in \mathcal{V}_{A \setminus \{a_i\}}} \cdot \mathbb{1}\left\{ \alpha = |\text{supp}(v)| \right\}$$
$$\cdot \mathbb{1}\left\{ t' = \text{top}_K\big(f(\mathcal{D}_{tr}[v; a_i \leftarrow 0]) \mid t_{val}\big) \right\} \cdot \mathbb{1}\left\{ t'' = \text{top}_K\big(f(\mathcal{D}_{tr}[v; a_i \leftarrow 1]) \mid t_{val}\big) \right\}$$
$$\cdot \mathbb{1}\left\{ \gamma' = \text{tally}\big(f(\mathcal{D}_{tr}[v; a_i \leftarrow 0]) \mid t', t_{val}\big) \right\} \cdot \mathbb{1}\left\{ \gamma'' = \text{tally}\big(f(\mathcal{D}_{tr}[v; a_i \leftarrow 1]) \mid t'', t_{val}\big) \right\}. \tag{25}$$

We can use the Boolean function $\phi(v \mid t_i, t_{val}, t', t'')$ in Equation 8 to simplify the above equation. Notice that the conditions $t' \in f(\mathcal{D}_{tr}[v; a_i \leftarrow 0])$ and $t'' \in f(\mathcal{D}_{tr}[v; a_i \leftarrow 1])$ are embedded in the definition of $\phi(v \mid t_i, t_{val}, t', t'')$ which will return $\infty$ if those conditions are not met. When the conditions are met, $\phi(v \mid t_i, t_{val}, t', t'')$ returns exactly the same triple $(\alpha, \gamma', \gamma'')$. Therefore it is safe to replace the five indicator functions in the above formula with a single one as such:

$$\omega(\alpha, \gamma', \gamma'' \mid t_i, t_{val}, t', t'') := \sum_{v \in \mathcal{V}_{A \setminus \{a_i\}}} \mathbb{1}\{(\alpha, \gamma', \gamma'') = \phi(v \mid t_i, t_{val}, t', t'')\} \tag{26}$$

Given our assumption that $\phi(v \mid t_i, t_{val}, t', t'')$ can be represented by an ADD with a root node $n(t_i, t_{val}, t', t'')$, the above formula is exactly the model counting operation:

$$\omega(\alpha, \gamma', \gamma'' \mid t_i, t_{val}, t', t'') := \text{count}_{(\alpha, \gamma', \gamma'')}(n(t_i, t_{val}, t', t'')) \tag{27}$$

$\square$

**Theorem E.1.** *If we can represent the Boolean function $\phi(v \mid t_i, t_{val}, t', t'')$ defined in Equation 8 with an Additive Decision Diagram of size polynomial in $|\mathcal{D}_{tr}|$ and $|f(\mathcal{D}_{tr})|$, then we can compute the counting oracle $\omega(\cdot \mid t_i, t_{val}, t', t'')$ in time polynomial in $|\mathcal{D}_{tr}|$ and $|f(\mathcal{D}_{tr})|$.*

*Proof.* This theorem follows from the two previously proved lemmas: Lemma E.1 and Lemma E.3. Namely, as a result of Lemma E.3 we claim that model counting of the Boolean function $\phi(v \mid t_i, t_{val}, t', t'')$ is equivalent to computing the oracle result. On top of that, as a result of Lemma E.1 we know that we can perform model counting in time linear in the size of the decision diagram. Hence, if our function $\phi(v \mid t_i, t_{val}, t', t'')$ can be represented with a decision diagram of size polynomial in the size of data, then we can conclude that computing the oracle result can be done in time polynomial in the size of data. $\square$

### E.3 DETAILS ON ADDITIVE MODEL QUALITY METRICS

**False Negative Rate** Apart from accuracy which represents a trivial example of an additive utility, we can show how some more complex utilities happen to be additive and can therefore be decomposed according to Equation 5. As an example, we use *false negative rate (FNR)* which can be defined as such:

$$m(\mathcal{D}_{tr}, \mathcal{D}_{val}) := \frac{\sum_{t_{val} \in f(\mathcal{D}_{val})} \mathbb{1}\{(\mathcal{A} \circ f(\mathcal{D}_{tr}))(t_{val}) = 0\} \mathbb{1}\{y(t_{val}) = 1\}}{|\{t_{val} \in \mathcal{D}_{val} : y(t_{val}) = 1\}|}. \tag{28}$$

In the above expression we can see that the denominator only depends on $\mathcal{D}_{val}$ which means it can be interpreted as the scaling factor $w$. We can easily see that the expression in the numerator neatly fits the structure of Equation 5 as long as we we define $m_T$ as $m_T(y_{pred}, (x_{val}, y_{val})) := \mathbb{1}\{y_{pred} = 0\}\mathbb{1}\{y_{val} = 1\}$. Similarly, we are able to easily represent various other utilities, including: false positive rate, true positive rate (i.e. recall), true negative rate (i.e. specificity), etc. We describe an additional example in subsection 3.3.

**Equalized Odds Difference**   We show how slightly more complex utilities can also be represented as additive, with a little approximation, similar to the one described above. We will demonstrate this using the "equalized odds difference" utility, a measure of (un)fairness commonly used in research Hardt et al. (2016); Barocas et al. (2019) that we also use in our experiments. It can be defined as such:

$$m(\mathcal{D}_{tr}, \mathcal{D}_{val}) := \max\{TPR_\Delta(\mathcal{D}_{tr}, \mathcal{D}_{val}), FPR_\Delta(\mathcal{D}_{tr}, \mathcal{D}_{val})\}. \tag{29}$$

Here, $TPR_\Delta$ and $FPR_\Delta$ are *true positive rate difference* and *false positive rate difference* respectively. We assume that each tuple $t_{tr} \in f(\mathcal{D}_{tr})$ and $t_{val} \in f(\mathcal{D}_{val})$ have some sensitive feature $g$ (e.g. ethnicity) with values taken from some finite set $\{G_1, G_2, ...\}$, that allows us to partition the dataset into *sensitive groups*. We can define $TPR_\Delta$ and $FPR_\Delta$ respectively as

$$
\begin{aligned}
TPR_\Delta(\mathcal{D}_{tr}, \mathcal{D}_{val}) &:= \max_{G_i \in G} TPR_{G_i}(\mathcal{D}_{tr}, \mathcal{D}_{val}) - \min_{G_j \in G} TPR_{G_j}(\mathcal{D}_{tr}, \mathcal{D}_{val}), \text{ and} \\
FPR_\Delta(\mathcal{D}_{tr}, \mathcal{D}_{val}) &:= \max_{G_i \in G} FPR_{G_i}(\mathcal{D}_{tr}, \mathcal{D}_{val}) - \min_{G_j \in G} FPR_{G_j}(\mathcal{D}_{tr}, \mathcal{D}_{val}).
\end{aligned} \tag{30}
$$

For some sensitive group $G_i$, we define $TPR_{G_i}$ and $FPR_{G_i}$ respectively as:

$$
\begin{aligned}
TPR_{G_i}(\mathcal{D}_{tr}, \mathcal{D}_{val}) &:= \frac{\sum_{t_{val} \in f(\mathcal{D}_{val})} \mathbb{1}\{(\mathcal{A} \circ f(\mathcal{D}_{tr}))(t_{val}) = 1\}\mathbb{1}\{y(t_{val}) = 1\}\mathbb{1}\{g(t_{val}) = G_i\}}{|\{t_{val} \in \mathcal{D}_{val} \ : \ y(t_{val}) = 1 \wedge g(t_{val}) = G_i\}|}, \text{ and} \\
FPR_{G_i}(\mathcal{D}_{tr}, \mathcal{D}_{val}) &:= \frac{\sum_{t_{val} \in f(\mathcal{D}_{val})} \mathbb{1}\{(\mathcal{A} \circ f(\mathcal{D}_{tr}))(t_{val}) = 1\}\mathbb{1}\{y(t_{val}) = 0\}\mathbb{1}\{g(t_{val}) = G_i\}}{|\{t_{val} \in \mathcal{D}_{val} \ : \ y(t_{val}) = 0 \wedge g(t_{val}) = G_i\}|}
\end{aligned}
$$

For a given training dataset $\mathcal{D}_{tr}$, we can determine Equation 29 whether $TPR_\Delta$ or $FPR_\Delta$ is going to be the dominant metric. Similarly, given that choice, we can determine a pair of sensitive groups $(G_{max}, G_{min})$ that would end up be selected as minimal and maximal in Equation 30. Similarly to the conversion shown in subsection 3.3, we can treat these two steps as a `reduce` operation over the whole dataset. Then, if we assume that this intermediate result will remain stable over subsets of $\mathcal{D}_{tr}$, we can approximatly represent the equalized odds difference utility as an additive utility.

As an example, let us assume that we have determined that $TPR_\Delta$ dominates over $FPR_\Delta$, and similarly that the pair of sensitive groups $(G_{max}, G_{min})$ will end up being selected in Equation 30. Then, our tuple-wise utility $u_T$ and the scaling factor $w$ become

$$
\begin{aligned}
m_T(y_{pred}, t_{val}) &:= TPR_{G_{max}, T}(y_{pred}, t_{val}) - TPR_{G_{min}, T}(y_{pred}, t_{val}), \\
w &:= 1/|\{t_{val} \in \mathcal{D}_{val} \ : \ y(t_{val}) = 1 \wedge g(t_{val}) = G_i\}|,
\end{aligned}
$$

where

$$TPR_{G_i, T}(y_{pred}, t_{val}) := \mathbb{1}\{y_{pred} = 1\}\mathbb{1}\{y(t_{val}) = 1\}\mathbb{1}\{g(t_{val}) = G_i\}.$$

A similar approach can be taken to define $m_T$ and $w$ for the case when $FPR_\Delta$ dominates over $TPR_\Delta$. Then, if we plug them into Equation 5, we obtain an approximate version of the equalized odds difference utility as defined in Equation 29. This approximation relies on the stability of the choices of $\min$ and $\max$ in Equation 30 and on the choice between $TPR$ and $FPR$ in Equation 29 (both of which can be precomputed).

# F   SPECIAL CASE: COMPUTING SHAPLEY FOR 1-NEAREST-NEIGHBOR CLASSIFIERS

We can significantly reduce the time complexity for 1-NN classifiers, an important special case of $K$-NN classifiers that is commonly used in practice. For each validation tuple $t_{val}$, there is always *exactly* one tuple that is most similar to $t_{val}$. Below we illustrate how to leverage this observation to construct the counting oracle. In the following, we assume that $a_i$ is the variable corresponding to the tuple for which we hope to compute the Shapley value.

Let $\phi_t$ represent the event when $t$ is the top-1 tuple:

$$\phi_t := p(t) \wedge \bigwedge_{\substack{t' \in f(\mathcal{D}_{tr}) \\ \sigma(t') > \sigma(t)}} \neg p(t'). \tag{31}$$

For Equation 31 to be *true* (i.e. for tuple $t$ to be the top-1), all tuples $t'$ where $\sigma(t') > \sigma(t)$ need to be *absent* from the pipeline output. Hence, for a given value assignment $v$, all provenance polynomials that control those tuples, i.e., $p(t')$, need to evaluate to false.

We now construct the event

$$\phi_{t,t'} := \phi_t[a_i/\text{false}] \wedge \phi_{t'}[a_i/\text{true}],$$

where $\phi_t[a_i/\text{false}]$ means to substitute all appearances of $a_i$ in $\phi_t$ to false. This event happens only if if $t$ is the top-1 tuple when $a_i$ is false and $t'$ is the top-1 tuple when $a_i$ is true. This corresponds to the condition that our counting oracle counts models for. Expanding $\phi_{t,t'}$, we obtain

$$\phi_{t,t'} := \left( p(t) \wedge \bigwedge_{\substack{t'' \in f(\mathcal{D}_{tr}) \\ \sigma(t'') > \sigma(t)}} \neg p(t'') \right)[a_i/\text{false}] \wedge \left( p(t') \wedge \bigwedge_{\substack{t'' \in f(\mathcal{D}_{tr}) \\ \sigma(t'') > \sigma(t')}} \neg p(t'') \right)[a_i/\text{true}]. \tag{32}$$

Note that $\phi_{t,t'}$ can only be *true* if $p(t')$ is true when $a_i$ is true and $\sigma(t) < \sigma(t')$. As a result, all provenance polynomials corresponding to tuples with a higher similarity score than that of $t$ need to evaluate to false. Therefore, the only polynomials that can be allowed to evaluate to true are those corresponding to tuples with lower similarity score than $t$. Based on these observations, we can express the counting oracle for different types of ML pipelines.

**Map Pipeline.** In a *map* pipeline, the provenance polynomial for each tuple $t'_i \in f(\mathcal{D}_{tr})$ is defined by a single distinct variable $a_i \in A$. Furthermore, from the definition of the counting oracle (Equation 7), we can see that $\omega(\cdot | t_i, t_{val}, t', t'')$ counts the value assignments that result in support size $\alpha$ and label tally vectors $\gamma'$ and $\gamma''$. Given our observation about the provenance polynomials that are allowed to be set to true, we can easily construct an expression for counting valid value assignments. Namely, we have to choose exactly $\alpha$ variables out of the set $\{t' \in f(\mathcal{D}_{tr}) : \sigma(t', t_{val}) < \sigma(t_i, t_{val})\}$, which corresponds to tuples with a lower similarity score than that of $t_i$ (measured by the similarity function $\sigma$). This can be constructed using a *binomial coefficient*. Furthermore, when $K = 1$, the label tally $\gamma'$ is entirely determined by the top-1 tuple $t'$. The same observation goes for $\gamma''$ and $t''$. To denote this, we define a constant $\Gamma_L$ parameterized by some label $L$. It represents a tally vector with all values 0 and only the value corresponding to label $L$ being set to 1. We thus need to fix $\gamma'$ to be equal to $\Gamma_{y(t)}$ (and the same for $\gamma''$). Finally, as we observed earlier, when computing $\omega(\cdot | t_i, t_{val}, t', t'')$ for $K = 1$, the provenance polynomial of the tuple $t''$ must equal $a_i$. With these notions, we can define the counting oracle as

$$\omega(\alpha, \gamma', \gamma'' | t_i, t_{val}, t', t'') = \binom{|\{t''' \in f(\mathcal{D}_{tr}) : \sigma(t''', t_{val}) < \sigma(t_i, t_{val})\}|}{\alpha} \mathbb{1}\{p(t'') = a_i\}$$
$$\mathbb{1}\{\gamma' = \Gamma_{y(t')}\}$$
$$\mathbb{1}\{\gamma'' = \Gamma_{y(t'')}\}. \tag{33}$$

Note that we always assume $\binom{a}{b} = 0$ for all $a < b$. Given this, we can prove the following corollary about *map* pipelines:

**Corollary F.1.** *For the 1-NN accuracy utility and a* map *pipeline, which takes as input a dataset of size $N$, the Shapley value can be computed in $O(N \log N)$ time.*

*Proof.* We start off by plugging in the oracle definition from Equation 33 into the Shapley value computation Equation 6:

$$\varphi(t_i, t_{val}) = \frac{1}{N} \sum_{t', t'' \in f(\mathcal{D}_{tr})} \sum_{\alpha=1}^{N} \binom{N-1}{\alpha}^{-1} \sum_{\gamma', \gamma'' \in \Gamma_{\mathcal{Y}, K}} m_\Delta(\gamma', \gamma'' | t_{val})$$
$$\binom{|\{t''' \in f(\mathcal{D}_{tr}) \; : \; \sigma(t''', t_{val}) < \sigma(t_i, t_{val})\}|}{\alpha}$$
$$\mathbb{1}\{p(t'') = a_i\}$$
$$\mathbb{1}\{\gamma' = \Gamma_{y(t')}\}$$
$$\mathbb{1}\{\gamma'' = \Gamma_{y(t'')}\} \tag{34}$$

As we can see, the oracle imposes hard constraints on the tuple $t''$ and tally vectors $\gamma'$ and $\gamma''$. We will replace the tally vectors with their respective constants and the tuple $t''$ we will denote as $t_i$ because it is the only tuple associated with $a_i$. Because of this, we can remove the sums that iterate over them:

$$\varphi(t_i, t_{val}) = \frac{1}{N} \sum_{t' \in f(\mathcal{D}_{tr})} \sum_{\alpha=1}^{N} \binom{N-1}{\alpha}^{-1} m_\Delta(\Gamma_{y(t')}, \Gamma_{y(t_i)} | t_{val}) \binom{|\{t''' \in f(\mathcal{D}_{tr}) \; : \; \sigma(t''', t_{val}) < \sigma(t_i, t_{val})\}|}{\alpha} \tag{35}$$

We could significantly simplify this equation by assuming the tuples in $f(\mathcal{D})$ are sorted by decreasing similarity. We then obtain:

$$\varphi(t_i, t_{val}) = \frac{1}{N} \sum_{j=i}^{N} \sum_{\alpha=1}^{N} \binom{N-1}{\alpha}^{-1} m_\Delta(\Gamma_{y(t_j)}, \Gamma_{y(t_i)} | t_{val}) \binom{N-j}{\alpha} \tag{36}$$

We shuffle the sums a little by multiplying $\frac{1}{N}$ with $\binom{N-1}{\alpha}^{-1}$ and we expand $m_\Delta$ based on its definition in subsection E.1. We also alter the limit of the innermost sum because $\alpha \le N - j$. Thus, we obtain:

$$\varphi(t_i, t_{val}) = \sum_{j=i}^{N} \Big( m_T(y(t_i), t_{val}) - m_T(y(t_j), t_{val}) \Big) \sum_{\alpha=1}^{N-j} \binom{N}{\alpha}^{-1} \binom{N-j}{\alpha} \tag{37}$$

The innermost sum in the above equation can be simplified by applying the so-called Hockey-stick identity Ross (1997). Specifically, $\binom{N}{\alpha}^{-1}\binom{N-j}{\alpha}$ becomes $\binom{N}{j}^{-1}\binom{N-\alpha}{j}$. Then, $\sum_{\alpha=1}^{N-j} \binom{N}{j}^{-1}\binom{N-\alpha}{j}$ becomes $\binom{N}{j}^{-1}\binom{N}{j+1}$. Finally, we obtain the following formula:

$$\varphi(t_i, t_{val}) = \sum_{j=i}^{N} \Big( m_T(y(t_i), t_{val}) - m_T(y(t_j), t_{val}) \Big) \binom{N-j}{j+1} \tag{38}$$

As we can see, the above formula can be computed in $O(N)$ iterations. Therefore, given that we still need to sort the dataset beforehand, the overall complexity of the entire Shapley value amounts to $O(N \log N)$. Here, we assume that we have a precomputed table of factorials from 1 to $N$ that allows us to compute the binomial coefficient in constant time. $\qquad \square$

**Computing the Shapley Value for the Entire Training Dataset.** Equation 38 represents a method for computing the Shapley value for a single data example $t_i$. When computing the Shapley value for every tuple in a training dataset, given that the tuples are sorted according to similarity to $t_{val}$, we can notice that the sum in Equation 38 exhibits the following recursive structure:

$$\varphi(t_i, t_{val}) = \varphi(t_{i+1}, t_{val}) + \Big( m_T(y(t_i), t_{val}) - m_T(y(t_j), t_{val}) \Big) \binom{N-i}{i+1}$$

If we take advantage of the above recursive structure, we can see that it is possible to compute the Shapley value for all data examples in a single pass that takes $O(N)$ time. Hence, since the overall

computation will still be dominated by the sorting procedure, the time to compute the Shapley value for all training tuples with respect to a single validation tuple $t_{val}$ is $O(N \log N)$.

**Fork Pipeline.** As we noted, both *map* and *fork* pipelines result in polynomials made up of only one variable. The difference is that in *map* pipeline each variable is associated with at most one polynomial, whereas in *fork* pipelines it can be associated with multiple polynomials. However, for 1-NN classifiers, this difference vanishes when it comes to Shapley value computation:

**Corollary F.2.** *For the* 1-*NN accuracy utility and a* fork *pipeline, which takes as input a dataset of size* $N$, *the Shapley value can be computed in* $O(N \log N)$ *time.*

*Proof.* We will prove this by reducing the problem of Shapley value computation in fork pipelines to the one of computing it for map pipelines. Let us have two tuples $t'_{j,1}, t'_{j,2} \in f(\mathcal{D})$, both associated with some variable $a_j \in A$. That means that $p(t'_{j,1}) = p(t'_{j,2})$. If we examine Equation 31, we notice that it will surely evaluate to false if either $\sigma(t'_{j,1}) > \sigma(t)$ or $\sigma(t'_{j,2}) > \sigma(t)$. The same observation holds for Equation 32.

Without loss of generality, assume $\sigma(t_{j,1}) > \sigma(t_{j,2})$. Then, $\sigma(t_{j,1}) > \sigma(t)$ implies $\sigma(t_{j,2}) > \sigma(t)$. As a result, we only ever need to check the former condition without paying attention to the latter. The outcome of this is that for all sets of tuples associated with the same variable, it is safe to ignore all of them except the one with the highest similarity score, and we will nevertheless obtain the same oracle result. Since we transformed the problem to one where for each variable we have to consider only a single associated tuple, we have effectively reduced the problem to the one of computing Shapley value for map pipelines. Consequently, we can apply the same algorithm and will end up with the same time complexity. $\square$

## G  DETAILS ABOUT THE EXPERIMENTAL PROTOCOL AND ADDITIONAL EVALUATION RESULTS

**Hardware and Platform.** All experiments were conducted on an AMD EPYC 7742 2.25GHz CPU. We ran each experiment in single-thread mode. All deep learning models were running on an NVIDIA A100 GPU.

**Datasets.** We assemble a collection of widely used datasets with diverse modalities (i.e. tabular, textual, and image datasets). Table 2 summarizes the datasets that we used. In each experiment, we subsample the dataset to $1K$ training data examples by using different random seeds.

Table 2: Datasets characteristics

| Dataset | Modality | # Examples | # Features | Label Noise |
|---|---|---|---|---|
| UCI Adult     (Kohavi et al., 1996) | tabular | $49K$ | 14 | injected |
| FolkUCI Adult     (Ding et al., 2021) | tabular | $1.6M$ | 10 | injected |
| FashionMNIST     (Xiao et al., 2017) | image | $14K$ | $28 \times 28$ | injected |
| 20NewsGroups     (Joachims, 1996) | text | $1.9K$ | $20K$ after TF-IDF | injected |
| DataPerf Vision(Mazumder et al., 2022) | tabular | 1.1 | 2048 | human error |
| CIFAR $-$ N     (Wei et al., 2022) | image | $50K$ | $32 \times 32 \times 3$ | human error |

**Models.** We use three downstream ML models following the previous feature extraction pipelines: XGBoost, LogisticRegression, and KNearestNeighbor. We use the `LogisticRegression` and `KNeighborsClassifier` provided by the `sklearn` package. We set `max_iter` to 5,000 for LogisticRegression and set `n_neighbors` to 1 for KNearestNeighbor.

**Protocol.** We conduct a series of experimental runs that simulate a real-world importance-driven data debugging workflow. In each experimental run, we select a dataset, pipeline, target model, and data repair method. If a dataset does not already have human-generated label errors, we follow the protocol of Li et al. (2021) and Jia et al. (2021) and artificially inject $50\%$ of label noise. Label noise injection is performed by selecting a random subset representing $50\%$ of training data examples, and replace the original label with some other valid label in a given dataset. Given the selected data repair method, we compute the importance using a *validation dataset*. We use this computed iportance to sort the training dataset. Data repairs will be conducted using this sorting order. If the repair method is random, the data is sorted randomly. We divide the range between $0\%$ data examined and $100\%$ data examined into 100 checkpoints. Specifically, at each checkpoint, we select the next batch out

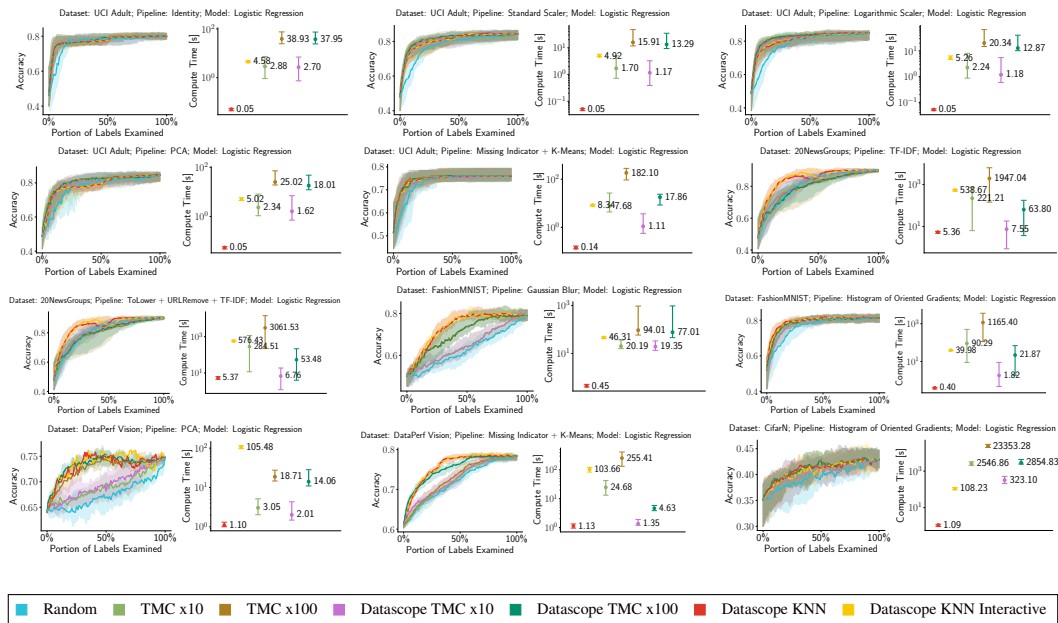

Figure 10: Label Repair experiment results over various combinations of datasets (1k samples) and map pipelines. We optimize for accuracy. The model is logistic regression.

of a 100 batches of data examples ordered based on the importance-based sorting order. We repair the labels in the given batch and we measure the quality of the given target model on a separate *test dataset* using some metric (e.g. accuracy). We also measure the time spent on computing importance scores. At any given checkpoint, the *label effort* represents the portion of data that was covered in all batches that were processed up to that checkpoint. We repeat each experiment 10 times with different random seeds and report the median as well as the 90-th percentile range (either shaded or with error bars).

## G.1    ADDITIONAL LABEL REPAIR EXPERIMENTS

We present the results of an array of experiments that were conducted for the label repair scenario. See section 5 for details on the experimental protocol. See Figure 10 to Figure 14 for experiments where we focus on improving accuracy. See Figure 15 to Figure 19 for experiments that explore the tradeoff between accuracy and fairness. Finally, in Figure 20 we show more results for the label repair experiments over deep learning embedding models for image and text data.

*Note about Fork Variants*: We create a "fork" version of the above pipelines, by prepending each with a `DataProvider` operator. It simulates distinct data providers, each providing a portion of the data. The original dataset is split into a given number of groups (100 in our experiments). We compute the importance of each group, and we conduct data repairs on entire groups all at once.

## G.2    ADDITIONAL SCALABILITY EXPERIMENTS

We provide results of additional experiments where we attempt to measure the trends of both the label repair efficiency and compute time, as a function of dataset size. To achieve this, instead of evaluating on synthetic data, we evaluate on CIFAR-N, a real-world dataset with human-generated label noise (Figure 23). We use logistic regression as a target model and the HOG transform pipeline for feature extraction. We keep the training and test set size to $5K$ data exaples and we vary the training set size from $1K$ to $32K$. We can notice that for training set of size $32K$, the TMC method requires around 1 day to complete with 10 Monte Carlo iterations and around 10 days with 100 iterations. At the same time we can notice that the KNN approximation is able to complete in a matter of minutes.

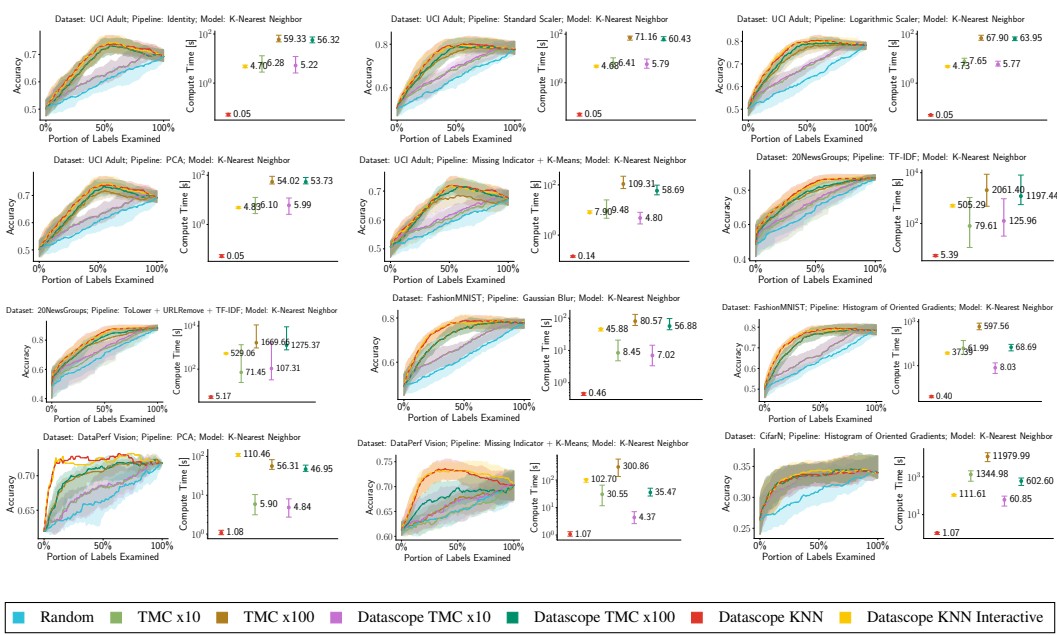

Figure 11: Label Repair experiment results over various combinations of datasets (1k samples) and map pipelines. We optimize for accuracy. The model is K-nearest neighbor.

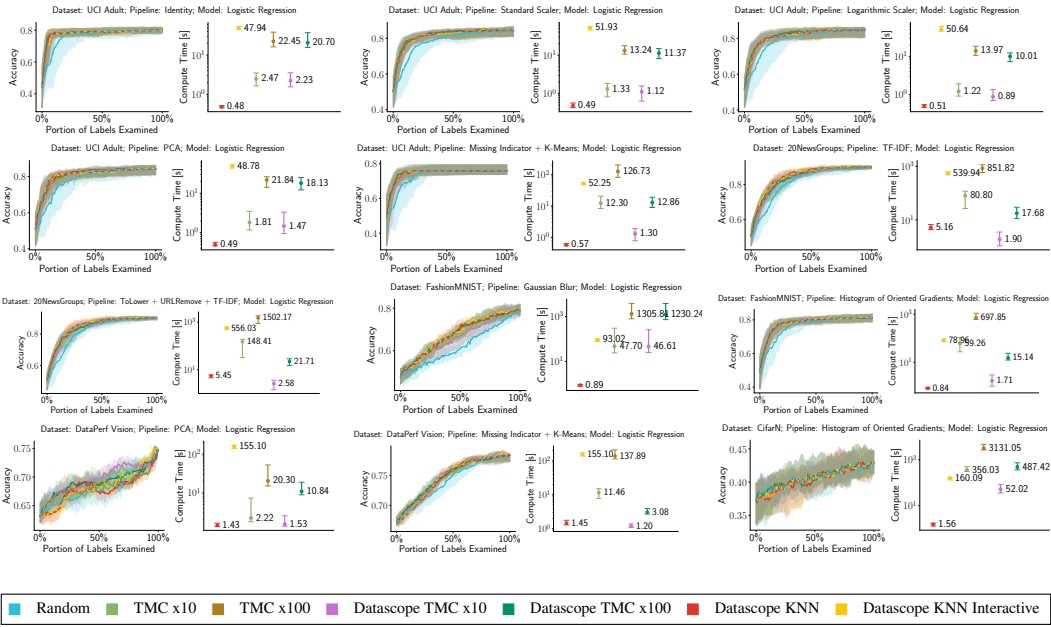

Figure 12: Label Repair experiment results over various combinations of datasets (1k samples) and fork pipelines. We optimize for accuracy. The model is logistic regression.

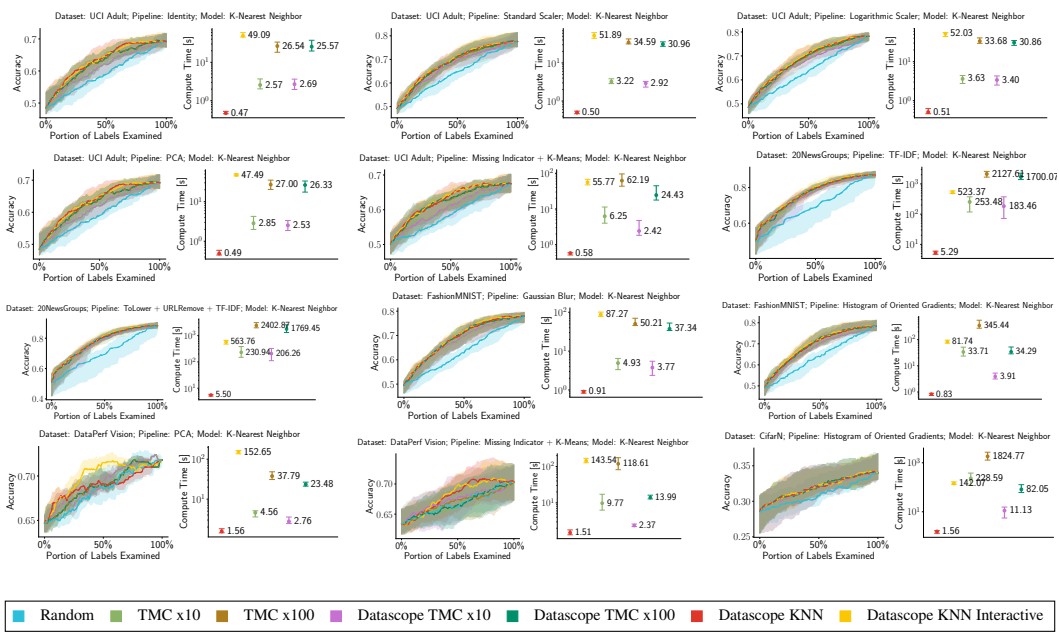

Figure 13: Label Repair experiment results over various combinations of datasets (1k samples) and fork pipelines. We optimize for accuracy. The model is K-nearest neighbor.

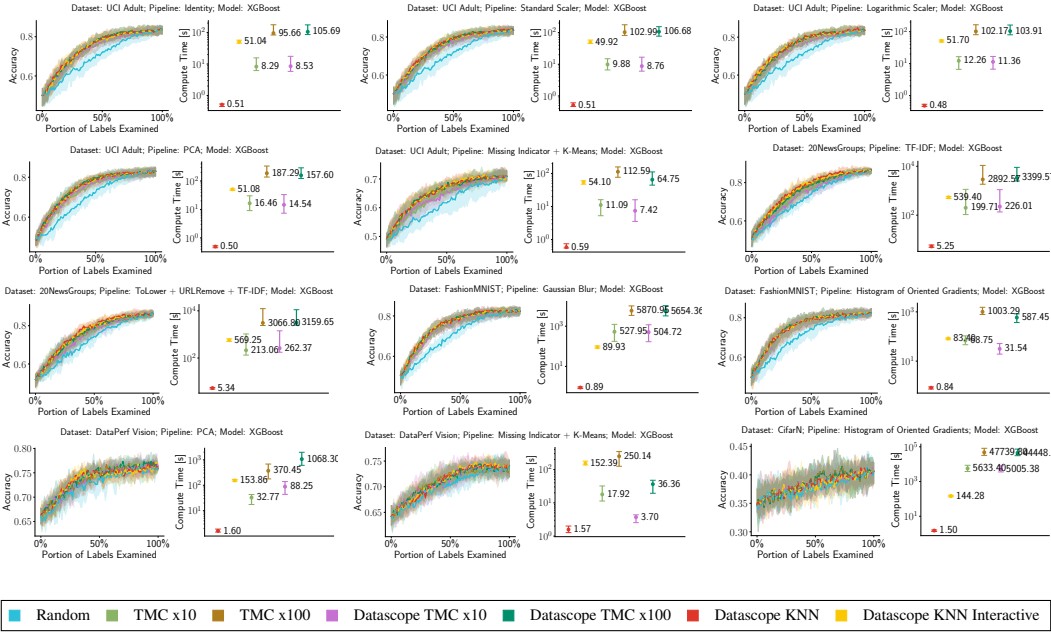

Figure 14: Label Repair experiment results over various combinations of datasets (1k samples) and fork pipelines. We optimize for accuracy. The model is XGBoost.

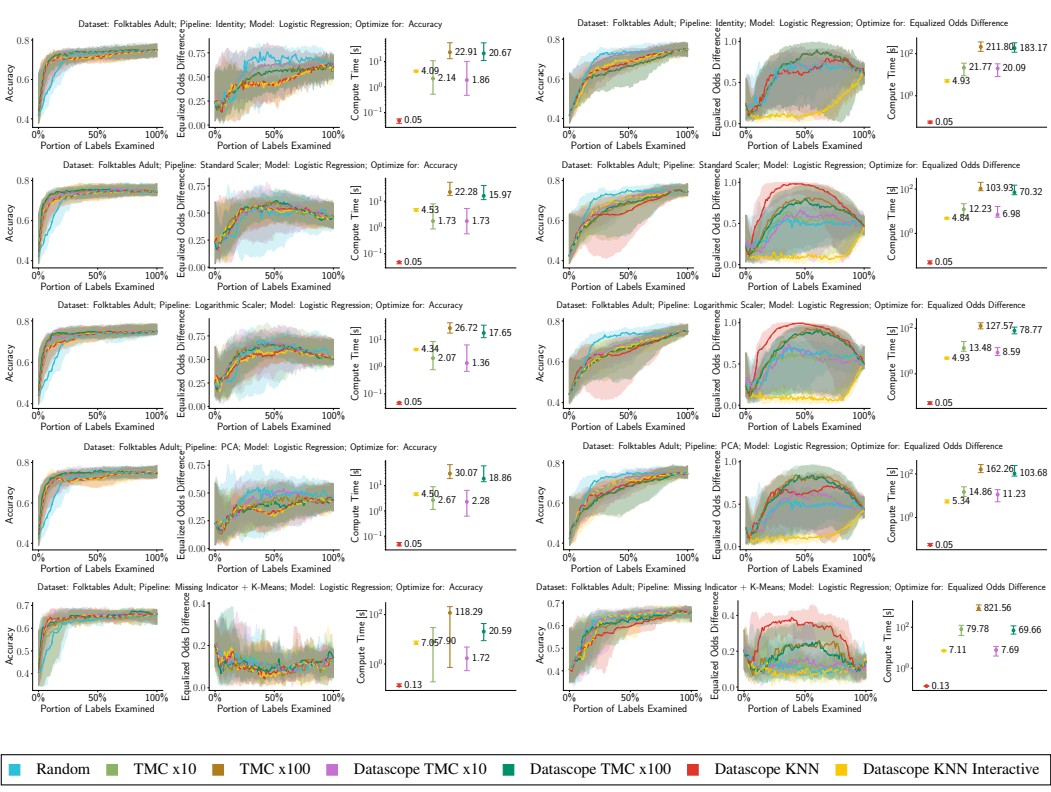

Figure 15: Label Repair experiment results over various combinations of datasets (1k samples) and map pipelines. We optimize for fairness. The model is logistic regression.

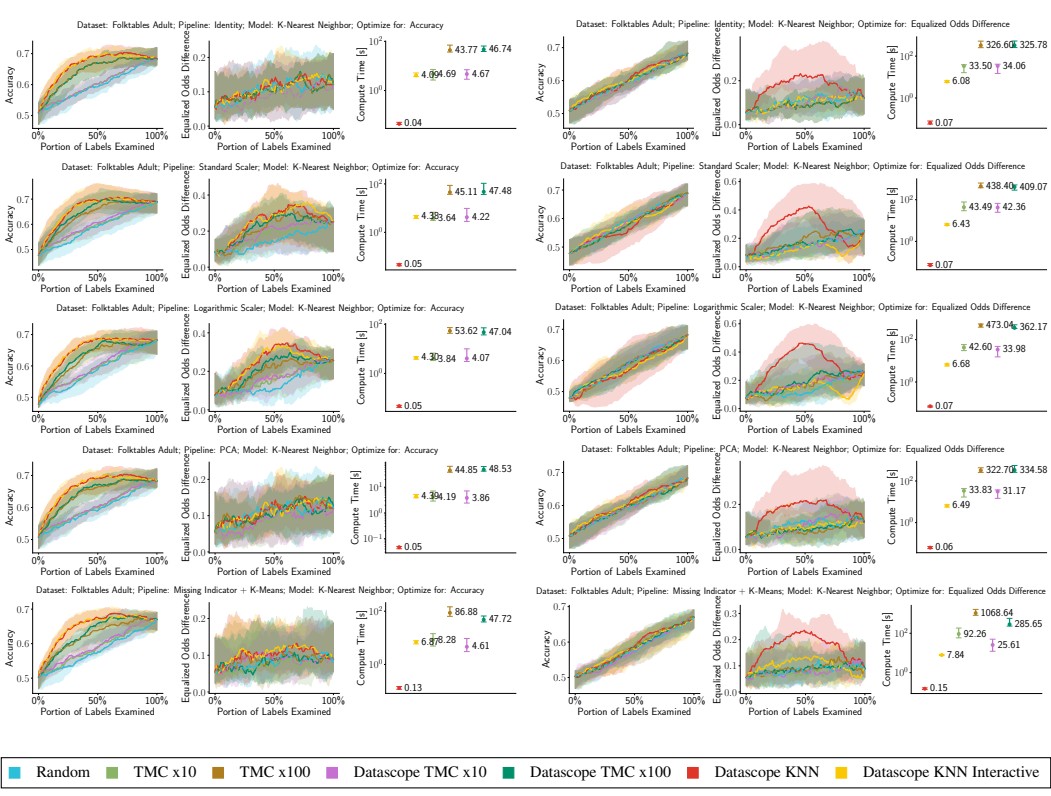

Figure 16: Label Repair experiment results over various combinations of datasets (1k samples) and map pipelines. We optimize for fairness. The model is K-nearest neighbor.

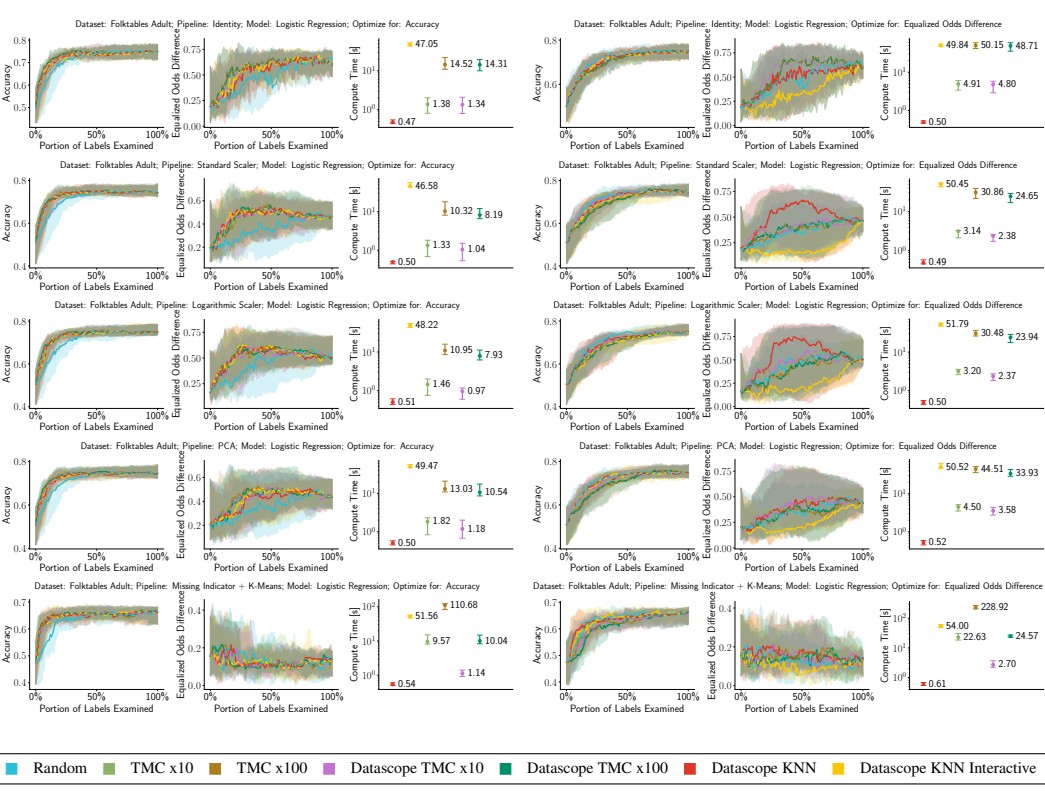

Figure 17: Label Repair experiment results over various combinations of datasets (1k samples) and fork pipelines. We optimize for fairness. The model is logistic regression.

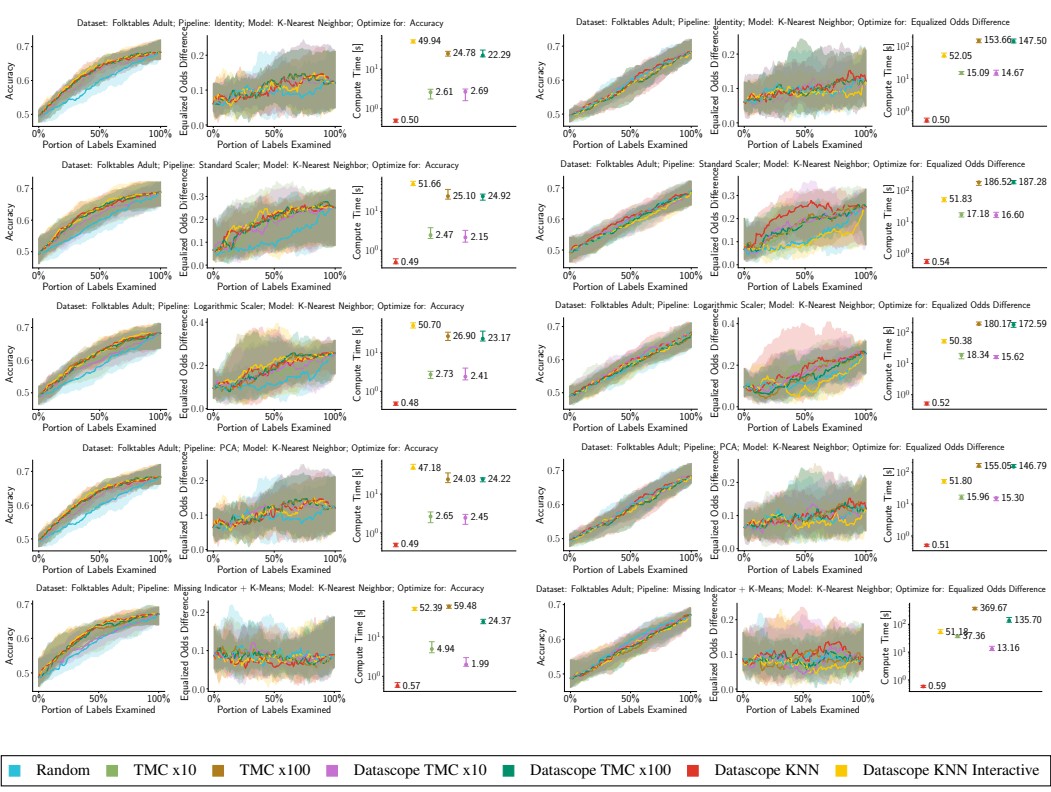

Figure 18: Label Repair experiment results over various combinations of datasets (1k samples) and fork pipelines. We optimize for fairness. The model is K-nearest neighbor.

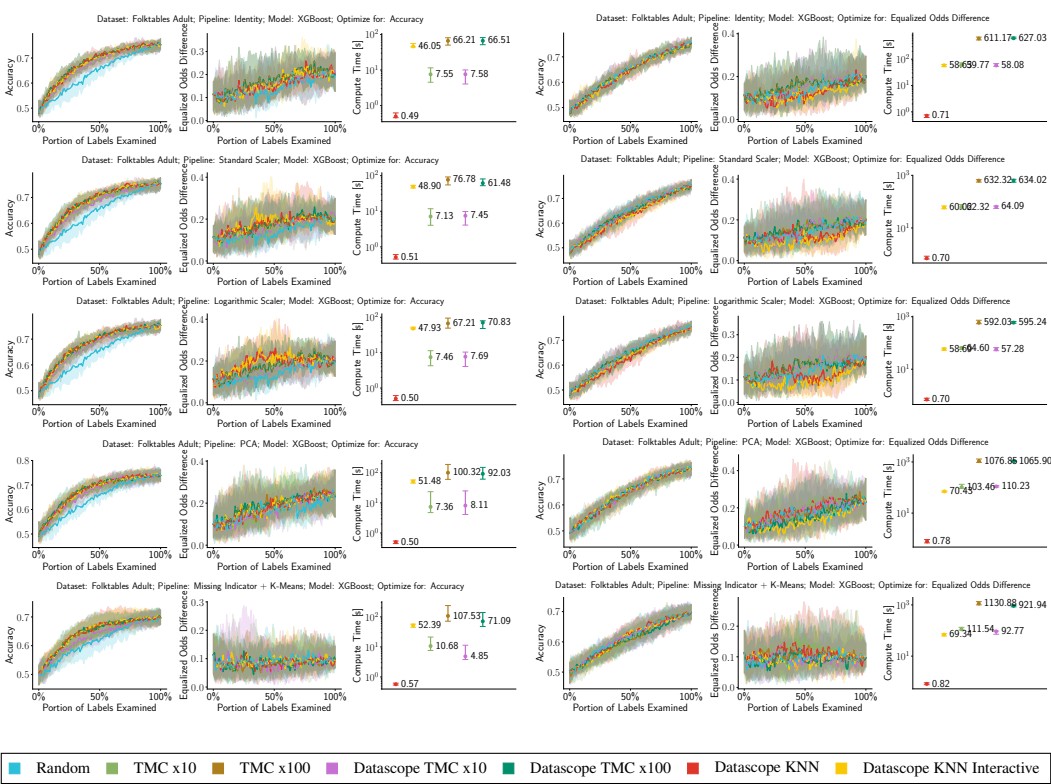

Figure 19: Label Repair experiment results over various combinations of datasets (1k samples) and fork pipelines. We optimize for fairness. The model is XGBoost.

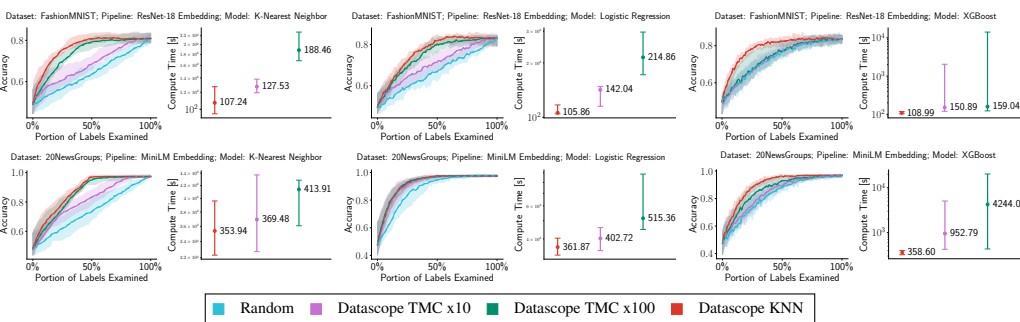

Figure 20: Label repair experiment executed over pipelines based on deep learning embedding models: ResNet-18 for image data, and the transformer based MiniLM for text data. Even though pipeline was executed on a GPU, this execution time dominates the overall importance compute times. Due to the long compute time of these pipelines we omit the vanilla black-box TMC methods.

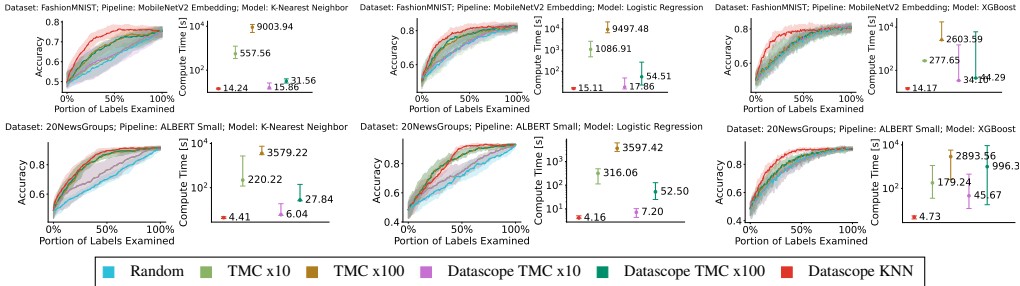

Figure 21: Label repair experiment executed over pipelines based on smaller deep learning embedding models. This permitted us to run both the Canonpipe TMC and vanilla TMC methods, along with our Canonpipe KNN method which still performs favorably compared to other baselines.

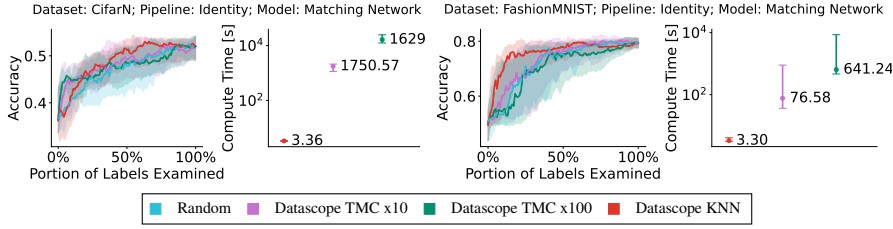

Figure 22: Experiments where we use matching networks, a one-shot learning approach, as a target model which we evaluate over the CifarN and FashionMNIST datasets.

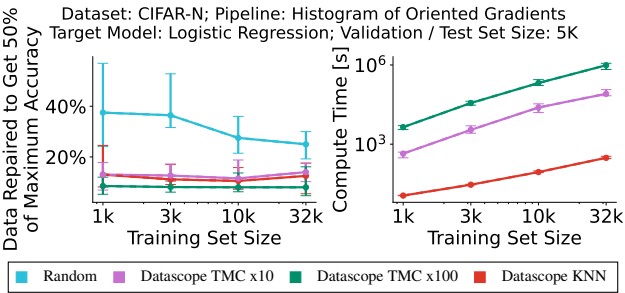

Figure 23: Evaluating how the label repair efficiency and compute time of Datascope scale as a function of dataset size. On the left-hand side we show how many data examples need to be repaired in order to recover $1/2$ of the maximum possible accuracy on the given dataset. We can notice that the KNN approximation is able to consistently achieve comparative label repair efficiency with orders of magnitude less compute time.

