# OpenReview forum: "Data Debugging with Shapley Importance over Machine Learning Pipelines"
_ICLR.cc/2024/Conference — ICLR 2024 poster_

### Official Review · Reviewer_5GXy · 2023-10-24

**Soundness:** 3 good
**Presentation:** 3 good
**Contribution:** 3 good
**Rating:** 8
**Confidence:** 3

**Summary:**

This paper proposes a method called Canonpipe for efficiently computing Shapley-based data importance over machine learning pipelines. The authors introduce several approximations that lead to significant speed-ups, making Canonpipe capable of data error discovery that is as effective as existing Monte Carlo baselines, and in some cases even outperform them. Overall, Canonpipe is a solution to the fundamental concern of discovering the data examples that are the most likely culprits for poor-quality ML models.

**Strengths:**

+ The authors tried to address a very important problem. i.e., computing Shapley value efficiently for a large dataset over general machine learning pipelines.
+ The authors proposed a novel provenance-based solution over canonical machine learning pipelines, which can address the computational challenges for evaluating Shapley values.
+ The authors also provided rigorous theoretical analysis for their proposed solution, which is sound and convincing
+ The authors also performed extensive experiments in different settings to demonstrate the effectiveness of the proposed method.

**Weaknesses:**

+ Some experimental details are missing, For example, in Section 5.2 when the authors discussed the scalability, I am not sure whether the results in Figure 7 are the ones for evaluating Shapley value for one sample or all the training samples. I guess it would be the former case. Otherwise, the time complexity would be quadratic. So it would be better if the authors could clarify this.
+ The overall experiments seem not to be comprehensive since the authors only evaluate their methods on simple dataset, e.g., FashinoMNIST, with simple models, such as KNN model. It would be great if the authors could demonstrate that their proposed method can handle large-scale datasets such as the ImageNet dataset
+ Although the authors refer to the earlier work on approximately evaluating Shapley values with KNN surrogate model, it is unclear to the readers how that can work, it would be better if the authors could briefly discuss this prior work, in particular what kind of correctness guarantee that can be obtained by approximately evaluating Shapley values with KNN surrogate models. This can make this paper more self-contained.
+ Also, the overall presentation could be improved. Although the authors mentioned that the PTIME computation time can be achieved with ADD. all the discussions of ADD are included in the appendix along with the main algorithm. Since this is the main contribution of this paper, it would be better if the authors could put some of these core technical parts in the main paper and briefly discuss them there.

**Questions:**

See above

---

> ### Author Response · Authors · 2023-11-22
> **Initial Comments - Part 1**
>
> We sincerely thank the reviewer for the appreciation they expressed for the problem we are trying to address, the novelty of our solution, as well as our attempt at a rigorous theoretical treatment and a thorough experimental evaluation. We also greatly appreciate the reviewer's constructive feedback. We provide our best attempt at addressing some concerns that were raised in the comments below (we apologize for the length) and in various edits we made to our paper draft (found in the revised version, marked with purple). Finally, in case the reviewer feels that their concerns are adequately addressed, we hope they would consider updating their final score accordingly.
>
> (W1) **I am not sure whether the results in Figure 7 are the ones for evaluating Shapley value for one sample or all the training samples**
>
> We thank the reviewer very much for this comment as they pointed out two flaws in our presentation that were unfortunately introducing some confusion. We address both flaws as follows:
>
> * In all our experiments, we always measure the compute time of Shapley value computation methods over the entire training dataset (as opposed to a single data example). We do this because we use importance scores to order the data repairs, so there is little use in computing data importance for a single data example. We apologize for never saying this explicitly in our original draft. We added a small modification clarifying this in the experimental protocol in section 5.1 (marked in purple).
> * The intuitive expectation of the reviewer that the curves in figure 7 should be quadratic is completely sensible, given the information we provide in our original draft. Unfortunately, we omitted a detail that reveals how we can leverage the recursive structure of the map pipeline Shapley formula in order to avoid re-running the entire computation for each training data example. We apologize for this mishap and we provide a brief clarification for this in section F of the appendix (marked in purple). The executive summary is that the structure of equation 38 can be leveraged to compute the Shapley value for all training data examples in a single pass. Hence the time complexity of computing the Shapley value for the entire training dataset for map pipelines remains $O(N \log N)$.
>
> (W2) **It would be great if the authors could demonstrate that their proposed method can handle large-scale datasets**
>
> This is another very good point that we try to address with an additional scalability experiment that we conducted and provide details for in section G.2 and figure 23 in the appendix (due to space limitations, marked in purple). Our goal with this experiment was to measure and observe trends of both the computation time and the data repair effectiveness as a function of dataset size. We measure effectiveness as the portion of labels that need to be repaired to close 50% of the gap between the accuracy with no data repairs and the accuracy with all data repaired. We selected the CIFAR-N as the dataset dataset (i.e. the CIFAR-10 dataset with human-generated label noise), the Histogram of Ordered Gradients as the preprocessing pipeline for extracting feature vectors from images, and Logistic Regression as the target model. We vary the size of the training dataset from 1K to 32K data examples, while we keep the validation and test dataset fixed to 5K data examples. We compare our Canonpipe KNN-approximation method with TMC-based methods. Due to severe slowdowns demonstrated by TMC methods for larger datasets, we do not include vanilla TMC, but only Canonpipe TMC which avoids re-running the preprocessing pipeline. We can observe that the KNN approximation is orders of magnitude faster than TMC. One striking observation is that for 32K training data examples, running Canonpipce TMC with 100 iterations took 10 days to compute the Shapley values. This reveals a glaring shortcoming of TMC methods when it comes to scalability. Since we wanted to run scalability experiments that compare the effectiveness of our KNN-based approximation to TMC baselines (which we use as the gold standard in the absence of exact Shapley values), we did not have time to scale the experiment beyond 32K data examples. Nevertheless, we hope that figure 23 is able to demonstrate the trend and give the reader a feeling of the effectiveness/efficiency trade-offs. Furthermore, in figure 7 where we focus on compute time only, we measure the time it takes to compute importance for up to 1M data examples. That said, given more time, we would happily run our method on the ImageNet dataset and, in case the paper is accepted we would provide those results in the camera-ready version.

---

> ### Author Response · Authors · 2023-11-22
> **Initial Comments - Part 2**
>
> (continued)
>
> (W3) **What kind of correctness guarantee can be obtained by approximately evaluating Shapley values with KNN surrogate models?**
>
> This is an excellent point that, to the best of our knowledge, targets a key shortcoming of the current state of the art. Unfortunately, we are not aware of any existing work that tries to explore theoretical *correctness guarantees* for computing the Shapley value using KNN surrogate models. There is prior work that addresses the original problem of KNN-based Shapley values without data preprocessing pipelines which is focused on computational efficiency (Jia et al. 2019b), as well as work that provides an empirical study of the effectiveness of applying the Shapley value to various data repair scenarios (Jia et al. 2021). At this point in time, we are only able to provide empirical arguments for the usefulness of KNN-based Shapley approximations. We believe that the question of theoretical guarantees is a fascinating one and we are eager to explore it in our future work. However, to us, this feels like quite a non-trivial pursuit that could likely turn into a whole new paper. In this paper we focus our scope on: (1) the efficiency of computation and establishing the relationship between computational complexity and pipeline structure; and (2) following the same line of empirical arguments as the current state of the art to demonstrate that there are many real-world settings where the KNN-based approximation works well, thus arguing for the usefulness of our approach. In section A of the appendix, we provide more details on the limitations of current KNN-based Shapley approximations, but this discussion is also focused only on computational complexity and not on correctness.
>
> (W4) **It would be better if the authors could put some of these core technical parts (e.g. ADDs) in the main paper and briefly discuss them there**
>
> We completely understand the point the reviewer is raising here! We also feel that ADDs are an interesting and novel aspect of our solution. Sadly, due to space limitations, we were forced to make hard decisions about what we were able to fit in the main body of the paper. For ADDs in particular, if we wanted to do them justice and present them in a comprehensible way, we feel that we would need at least an additional page, which we couldn't afford. Our intentions for the structure of the main body were the following: (1) introduce and motivate the problem; (2) introduce the concept of canonical pipelines and how we could analyze them using data provenance; (3) present key high-level components of our KNN-based algorithm; (4) provide a solid empirical evaluation of our method. We left all other details in the appendix with the hope that readers who are interested would appreciate a presentation of all components of our solution (including ADDs) that is more thorough and is not constrained by the number of pages.

---

> > ### Comment · Reviewer_5GXy · 2023-11-23
> >
> > Thanks for the authors' clarification! It makes more sense now. I would love to increase my score.

---

### Official Review · Reviewer_bts9 · 2023-10-28

**Soundness:** 3 good
**Presentation:** 3 good
**Contribution:** 3 good
**Rating:** 8
**Confidence:** 2

**Summary:**

This work studies a novel and relevant problem of incorporating Shapley-based data evaluation into data processing pipelines. The work first clarifies the current limitations on implementing Shapley methods with data processing pipelines–the Monte-Carlo sampling approach would necessitate re-running the data processing pipeline which costs significant time; KNN-based approximations are incompatible with some constraints posed by the data processing pipelines and thus sometimes cannot be applied. Then, this work proposes the concept of “canonical” pipelines which allow directly relating inputs and outputs. By approximating pipelines as canonical, the proposed methods may achieve significant speed-ups for the Monte Carlo approach. Also, by combining canonical pipelines with the K-nearest neighbor as a proxy model, the proposed  PTIME Shapley computation algorithms allow applying KNN Shapley as a special case applicable to map pipelines. The paper is technically solid.

**Strengths:**

The paper is clear, sharp, and well-structured. The paper is well-written, well-contextualized, and well-motivated. The language is technically sound. The identified problem is valid and important. The proposed technical approaches are well-documented with rigorous elaborations. This work could be of lots of interest to data science practitioners. Proposed methods achieve significant speedups in empirical studies.

**Weaknesses:**

I do not see major weaknesses. I'm familiar with the literature on Shapley methods and their practical implementations but not much on data processing pipelines in the real world. The review provided is limited by the scope of my knowledge. I would leave it to other reviewers to evaluate the practicalness of the modeling and treatment of the data processing pipelines.

- Format: Appendix is not cut from the main paper. The PDF provided for the main paper is this 34-page document.

**Questions:**

It would be nice if the authors could better contextualize the proposed framework with real-world applications, like, providing some concrete examples or a motivating case to help the audience better delve into the problem.

- Appendix should not be submitted under the main paper.

---

> ### Author Response · Authors · 2023-11-23
> **Initial Comments**
>
> We cannot thank the reviewer enough for highlighting so many positive aspects of our work! The summary provided hits all the major points of our contributions and gives us the impression that the confidence rating that the reviewer specified might be underestimating their true competencies. We provide a few brief comments below regarding several points raised by the reviewer.
>
> (W1) **Format: Appendix is not cut from the main paper. The PDF provided for the main paper is this 34-page document**
>
> We apologize if the reviewer was inconvenienced by the large document. Our decision to not split the paper in two is driven by the following reasons:
>
> * The ICLR Author Guide (link: https://iclr.cc/Conferences/2024/AuthorGuide) where the FAQ states that: (Q) Should the appendices be added as a separate PDF or in the same PDF as the main paper? (A) Either is allowed: you can include the appendices at the end of the main pdf after the references, or you can include it as a separate file for the supplementary materials.
> * Since we reference the appendix many times in the main body of the paper with clickable links, we thought the readers would have an easier time navigating our paper (as opposed to having to go back and forth between two documents).
>
> (Q1) **It would be nice if the authors could better contextualize the proposed framework with real-world applications**
>
> We thank the reviewer for raising this concern. We added a short example at the beginning of the introduction (marked in teal color) that hopefully introduces the reader to a tangible real-world application. Furthermore, we hope that figure 1 already gives the reader an overview of a real-world example of repairing tabular data. Finally, if the paper is accepted, we would publish our code on GitHub along with several real-world examples which would hopefully provide those who are interested with sufficient context.

---

> > ### Comment · Reviewer_bts9 · 2023-11-23
> >
> > Thanks to the authors for the rebuttal. I would keep my score in support of this work.

---

### Official Review · Reviewer_guJp · 2023-10-29

**Soundness:** 4 excellent
**Presentation:** 3 good
**Contribution:** 3 good
**Rating:** 6
**Confidence:** 3

**Summary:**

Data repair is typically performed on preprocessed data at the stage immediately preceding model training. This paper explores data valuation on raw data before preprocessing steps are performed. This necessitates a framework of data provenance in ML pipelines and a computation approach for data Shapley under a KNN approximation. The paper demonstrates the usefulness of this data valuation framework as achieving competitive performance in data debugging at significant compute cost/time reduction.

**Strengths:**

The paper is thorough in introducing a new problem setting within data valuation and debugging. The methodology builds on KNN Shapley but factors arbitrary general data preprocessing pipelines. The algorithm includes theoretical guarantees on polynomial time computation and an extensive set of experiments to demonstrate the compute efficiency of the proposed method.

**Weaknesses:**

- The framework depends on a data preprocessing pipeline to be the same for both training and validation (equation 2). However, one challenge of data valuation before pre-processing is that there may be different pre-processing pipelines between training and validation. For example, we can consider random data augmentation techniques used in training, but not for testing. Does this methodology handle different pre-processing pipelines, or for example, a validation pipeline that is a subset of the training pipeline?
- Near Section 3.2/3.3 (or a detailed version in the Appendix), it would be useful to have a detailed dictionary of common pipelines / components and how they fit into Map, Fork, Join, or can be approximated by Map-reduce. This could be a table for example similar to Table 1. A table such as this would make the significance of the proposed work more clear in terms of how an ML practitioner can think of pre-processing steps in these pipelines.
- If I understand the experiments correctly, the baselines are performing data importance on the raw data before pre-preprocessing. Furthermore, existing KNN Shapley methods cannot accurately model the combinatorial explosion in subsets obtained from a data pipeline, making them conceptually unattractive. However, one baseline in data repair may be to perform valuation on data points after pre-processing, and then simply invert the pipeline manually to determine the relevant raw data points. How would existing methods including KNN Shapley perform on label repair in terms of accuracy improvement and compute time? More generally, what is the practical significance of identifying data points for repair with a method that captures the pre-processing operations versus simply identifying potential points for repair after pre-processing and then inverting the preprocessing pipeline to determine affected raw data points?
- The experiment protocol is not thoroughly explained and relies on referencing prior work (e.g., noise injection, measuring labor cost). It would be useful to include this discussion perhaps in Appendix.
- There is some minor writing improvements to be made, for example, In page 2, set S is used without definition

**Questions:**

See weaknesses

---

> ### Author Response · Authors · 2023-11-22
> **Initial Comments - Part 1**
>
> We express sincere gratitude to the reviewer for providing valuable feedback. The reviewer raised several interesting points which we address below. We also provide corresponding changes made in the paper draft itself (marked in orange). We hope that the reviewer will consider updating their review score in case they find our comments and edits satisfying. Finally, we apologize for the length of our comments, but we were aiming to bring as much clarity as we could when responding to all the excellent points raised by the reviewer.
>
> (W1) **The framework depends on a data preprocessing pipeline to be the same for both training and validation**
>
> We thank the reviewer for the insightful observation. We would like to draw several points that hopefully bring more clarity and address the reviewer's concerns:
>
> * It is true that the way we present the pipeline $f$ being applied in the same manner to both the training and the validation datasets ignores the possibility of these pipelines being different. This is just an assumption we made to simplify our presentation.
> * The only natural constraint in this context is that the data format of both the processed training data and the processed validation data (and for that matter the processed test data as well) should be such that all of them are readable by the model. This was the main intention behind our decision to write that $f$ is applied to both the training and validation data, i.e. to signal that we are not passing raw validation data to the model.
> * It is an excellent observation that the data augmentation operator is not applied to validation and test data. Indeed, in our experiments, we apply also data augmentation to training data only.
> * Since we do not attempt or track provenance over validation or test data, the fact that the pipeline applied to those datasets might in fact differ from the one applied to training data does not contradict our subsequent analysis and conclusions.
>
> We added an explicit clarification on this topic in section 2 (marked in orange).
>
> (W2) **It would be useful to have a detailed dictionary of common pipelines/components and how they fit into Map, Fork, Join**
>
> We thank the reviewer for drawing our attention to this shortcoming of our presentation. We can see how our taxonomy of pipelines could indeed introduce confusion and would benefit from additional clarification. When thinking about how to address this, we felt that fitting it into a table might still be not clear enough without appropriate context. Hence, instead of a table, we provide a semi-structured taxonomy of different pipeline types (along with example operators) in section B of the appendix (marked in orange).

---

> ### Author Response · Authors · 2023-11-22
> **Initial Comments - Part 2**
>
> (continued)
>
> (W3) **On inverting the pipeline manually as a baseline approach / on the significance of a method that captures data preprocessing**
>
> It looks like this point has several related but distinct questions raised so we would like to address them separately.
>
> * **How do the baselines handle data preprocessing pipelines?** - The scenario we present in the paper always assumes importance being computed for raw data (i.e. before preprocessing; we argue below why this scenario is significant). To make existing baselines (e.g. Truncated Monte Carlo) handle this scenario, in each Monte Carlo iteration we assume that we need to re-run the entire pipeline for every training data subset that is sampled (note that MC methods avoid the combinatorial explosion by sampling the set of data subsets). Methods that we introduce in this paper (i.e. Canonpipe TMC and Canonpipe KNN) also compute the importance of raw data, but introduce approximations to save on computational time. Hence, all methods discussed in this paper compute the importance of raw data.
> * **Couldn't we have a baseline that simply inverts the pipeline manually to determine the relevant raw data points?** - We feel that this might be trickier to achieve than it might seem at first glance. Of course, for simple map pipelines, it is trivial to link output data examples to their corresponding raw data examples and perform data repairs. However, more complex pipelines introduce challenges. For example, if we consider a data augmentation (i.e. fork) pipeline which transforms each input tuple $t\_i$ to 10 output tuples $t\_{i,1}, ..., t\_{i,10}$. Then if we compute importance (i.e. Shapley values) for these output tuples, how should this be linked to input tuples? Let's say we have two tuples $t\_{i}$ and $t\_{j}$, and $t\_j$ should be repaired first. However, if we were to sort output tuples by importance, maybe $t\_{i, 1}$ has higher importance than $t\_{j, 1}$, but this could be because $t\_{i, 1}$ dominates over $t\_{i,2}, ..., t\_{i,10}$ while tuples coming from with $t\_j$ have similar importance. This would mean that we would repair $t\_i$ before $t\_j$ which should not happen. It is unclear to us how to prevent such situations. As another example, let's consider joining two sets of tuples and every output tuple is the product of merging two source tuples. If we compute the importance of the output tuple, how should this be propagated to the input tuples? Furthermore, in a one-to-many join pipeline, a tuple from the first set can be joined with multiple tuples from the second set, and result in multiple output tuples being linked to it. How should importance be propagated in this case? In this paper, we argue that handling such situations "the right way" requires us to come up with more principled approaches. We try to demonstrate that taking advantage of the data provenance framework is a fundamental approach to analyzing ML pipelines and should be regarded as a crucial component of future ML systems. Hence, in our experiments, we decided to focus on baselines that are aligned with a principled treatment of ML pipelines: (1) a black-box approach exemplified in the vanilla TMC; and (2) a white-box approach exemplified with Canonpipe TMC and Canonpipe KNN.
> * **What is the practical significance of identifying data points for repair with a method that captures the pre-processing?** - Our understanding of this question is that it asks why is it even needed to compute the importance of raw data as opposed to simply relying on existing methods for computing importance of data as it is fed into the ML model. As we argue in the introduction and depict in Figure 1, key observations that motivate our work are that: (1) data preprocessing pipelines are ubiquitous in real-world ML workflows; and (2) data errors (and subsequent data repairs) typically occur in raw unprocessed data. We argue that the fact that existing methods do not integrate data preprocessing pipelines into their analysis is a fundamental shortcoming, which we attempt to fix with our contributions.
>
> (W4) **The experiment protocol is not thoroughly explained and relies on referencing prior work**
>
> We apologize for the lack of clarity in our original experimental protocol. We are grateful for the reviewer's understanding that the space limitation forced us to leave many details out. We provide an extended version of the experimental protocol in section G of the appendix (marked in orange) which hopefully clarifies the key open questions.
>
> (W5) **In page 2, set S is used without definition**
>
> This was a minor mishap which we corrected in the revised version of section 2 (marked in orange). The symbol $S$ should have been replaced with $\mathcal{D}$ (to be consistent with equation 1).

---

### Official Review · Reviewer_V8Vh · 2023-11-02

**Soundness:** 2 fair
**Presentation:** 2 fair
**Contribution:** 3 good
**Rating:** 6
**Confidence:** 4

**Summary:**

This paper introduces a framework for identifying influential training examples in a machine learning pipeline using Shapley Values algorithm in an efficient manner. The first efficiency problem that the paper addresses is identifying whether the output of data preprocessing pipeline for a given training example belongs to a given subset of training examples in O(1) time. The paper claims that this condition holds for the following three pipelines: map, fork, and one-to-many join.
The second efficiency problem that the paper addresses is related to the performance of the ML model and the utility metric used to measure the quality of the model. In this case, the paper suggests to use KNN algorithm and requires that the model quality metric is additive. The authors show that their framework is computationally more performant compared to baseline approaches. It also reaches competitive results in terms of accuracy on downstream tasks.

**Strengths:**

1) The work discusses related work thoroughly and highlights their pros and cons and the relation to the work that they are proposing.
2) The visual figures 1 and 2 are well made and help with the understanding of the proposed method.
3) The work provides experimental results for a variety of different pipelines for text, tabular and image datasets.  It also shows improvements not only for runtime but also for accuracy and fairness metrics.

**Weaknesses:**

1) From the introduction and abstract of the paper there is an impression that the paper aims to identify influential training examples but they doesn't seem to be any experimental results on that aspect. The experimental results are mostly cumulative w.r.t. overall accuracy, runtime, etc. There are no examples that show the effectiveness of the method w.r.t. specific training instances.
2) To make the paper more clear it would be good to define what exactly “canonical” pipeline and “ data provenance”  mean in the beginning of the paper. The readers need to have a clear understanding of those terms.
3) The notation  `D_{tr}[v] to denote D` is a bit confusing.  t \in f(D_{tr})  is confusing too since t \in D_{tr} and we see exactly t \in D_{tr}  notation later in the paper. It would be good to change the notation to make it more straightforward.
4) In section 3.3  f* doesn’t seem to be defined too ?
5) Figure 3 is referenced in pages 4 and 5 and it is not explained. It’s unclear why the Compute time of Canonpipe TMC x100 is worse than TMC x10.
6) The intuitions behind modified KNN  and quality metrics in section 4.1 are unclear.
7) The description of  Counting Oracle is not very clear. It would be good, if possible, to describe it in a more intuitive way. It seems to be overloaded with math notations and is not straightforward to follow.

**Questions:**

1) Sections 3.2 - 3.3: why are  One-to-many, fork and join canonical ? How is the canonical pipeline defined ? Why exactly is `reduce` non-canonical ? Would you, please, bring examples ?
2) Are modified KNN  and quality metrics based on previous work or something new that the authors propose ?
3) What are the limitations of the work ?
4) Since we are using modified quality metric and ML algorithm (KNN), I wonder how practical is the approach in terms of non-KNN models and different quality metrics?

---

> ### Author Response · Authors · 2023-11-22
> **Initial Comments - Part 1**
>
> We are deeply thankful to the reviewer for their insightful feedback and we are grateful for their appreciation of our figures and experiments!
>
> We made our best effort to integrate their feedback which can be found in the provided revised version (changes marked in blue). Furthermore, we provide some comments below on some specific points that the reviewer thoughtfully drew our attention to. We apologize in advance for the length of our comments, but given the detailed review, we were compelled to address every single point. We sincerely hope that the reviewer finds our comments helpful in assessing our work. Finally, if the reviewer has any remaining concerns that prevent them from voting to accept our paper, we would eagerly invite them to raise them and give us the opportunity to provide further clarifications.
>
> (W1) **Showing the effectiveness of the method w.r.t. specific training instances**
>
> As the reviewer noted, in our experiments we rely on measuring model quality as a proxy to determine whether our data repairs are effective or not. The reasons for this are twofold: (1) The main goal of data repair in the context targeted by our work is to improve the model quality. Hence, we believe that the effectiveness of all data repairs should be judged based on how close they reach us to higher-quality models (w.r.t. to the specific quality measure we choose). (2) If we make interventions on data (e.g. label repairs) and we observe an improvement in model quality (e.g. accuracy), then it should be fair to establish a causal relationship between the two.
>
> That being said, we thought that the reviewer made a very interesting point and we made adjustments to Figure 3 and Figure 4 to include not only the observed accuracy but also the proportion of dirty labels that were correctly identified during the data repair process. That way the reader can observe the relationship between the discovery of dirty labels and the resulting change in model quality.
>
> (W2) **Define what exactly “canonical” pipeline and “ data provenance” mean in the beginning of the paper**
>
> We thank the reviewer for this request which will indeed improve the flow of the paper. The corresponding modifications can be found in the Contributions section on Page 2 (marked in blue).
>
> (W3) **It would be good to change the notation to make it more straightforward**
>
> We made many pases to the notation prior to submitting this paper in an attempt to strike a decent trade-off between clarity and completeness. We would love to be able to substantially simplify the notation, but we were afraid to not jeopardize the completeness of our proofs. Furthermore, since in this paper we are heavily reliant on techniques from data management theory, we had to borrow a lot of their notational practices, which might seem less familiar to ML folks. For example, using the $\mathcal{D}\_{tr}[v]$ notation to denote subsets of $\mathcal{D}\_{tr}$ is essential because we need to be able to uniquely identify all distinct subsets using value assignments $v$ in order to be able to integrate this with our model counting techniques. Furthermore, since we believe that the target audience of this paper is people who are interested in building efficient ML systems, and since we expect that those people are likely more familiar with the field of data management, we hope that they would find this notation slightly less confusing.
>
> That being said, we appreciate the feedback very much and in an effort to address it as much as we could, we made another pass which, hopefully, improved the consistency of our notation. Specifically, we made sure that the symbol $t$ (without the apostrophe) is always used for tuples from $\mathcal{D}\_{tr}$, while we add the apostrophe (e.g. $t'$) to tuples that represent the pipeline output $f(\mathcal{D}\_{tr})$. We marked all such changes with blue text. If the reviewer could point us to any more specific ways to improve our notation, we would happily apply them in the next revised version.
>
> (W4) **In section 3.3 f\* doesn’t seem to be defined**
>
> We thank the reviewer for pointing this out and we made the corresponding change (marked in blue).
>
> (W5) **It’s unclear why the Compute time of Canonpipe TMC x100 is worse than TMC x10**
>
> As mentioned in Section 5.1 (experimental setup), x10 and x100 refer to the number of Monte Carlo iterations. Even though Canonpipe TMC speeds up computation, doing fewer Monte Carlo iterations can often result in a faster compute time. We felt that the observation about Canonpipe TMC x100 being worse than TMC x10 was not too surprising so we did not want to draw attention to it as we were afraid to detract from the main message, which is that Canonpipe TMC is significantly faster than vanilla TMC (with the same number of Monte Carlo iterations). Nevertheless, we made a few changes in section 3.3 (marked in blue) which will hopefully clarify the meaning of x10 and x100 and help the reader interpret the figures better.

---

> ### Author Response · Authors · 2023-11-22
> **Initial Comments - Part 2**
>
> (continued)
>
> (W6) **The intuitions behind modified KNN and quality metrics in section 4.1 are unclear**
>
> We understand this comment to mean that the KNN model and additive quality metrics are introduced too abruptly without explaining why we are introducing them in the first place. We add a brief motivational paragraph at the beginning of section 4.1 (marked in blue).
>
> (W7) **It would be good, if possible, to describe counting oracles in a more intuitive way**
>
> We apologize for the density of section 4.2. Our goal was to introduce key building blocks of our method which are then applied in our analysis. However, due to lack of space, the presentation is perhaps a bit denser than ideal. In order to improve the reader's intuition, we added a brief clarification after equation 7 (marked in blue).
>
> (Q1) **How is the canonical pipeline defined? Why exactly is reduce non-canonical?**
>
> In order to address W2, we modified the contribution section on page 2 with a simple definition of canonical pipelines -- "We introduce the notion of a canonical pipeline which we simply define as a distinct pipeline structure that lends itself to efficiently relating pipeline inputs and outputs, as well as efficiently computing Shapley values." In other words, we just noticed some distinct types of operators, we group them based on how they impact the provenance polynomials and we name them based on the type of operation they perform. Then, if a pipeline lends itself to efficiently computing the Shapley value, then we call it "canonical". It is a loose concept meant to facilitate studying of different types of pipelines and their impact on the computability of Shapley values.
>
> The types of pipelines mentioned in this paper, along with the corresponding polynomial structure are: (1) map pipelines - the output tuples are associated with single-variable polynomials and each tuple has a distinct variable; (2) fork pipelines - the output tuples are also associated with single-variable polynomials, but multiple output tuples can share the same variable; (3) one-to-many join pipelines - the output tuples are a product of multiple variables, where each variable comes from one of the tuples that were matched and merged together in order to produce the output tuple; (4) reduce pipelines - since each output tuple depends on every single input tuple, reduce pipelines are tricky to model using the data provenance framework, since removing any tuple from the input set effectively changes every single tuple in the output set.
>
> We provide examples of real pipeline operators and categorize them in section B of the appendix (marked in orange). Furthermore, figure 2 contains a depiction of map, fork and join pipelines. To give an example of a reduce pipeline, consider a dataset with 5 tuples $t_1, ..., t_5$ and each one is associated with a single variable $a_1, ..., a_5$. Then, let's consider a sum operator which is a simple reduce operation that takes the 5 tuples and sums them up. However, in order to model the ability to add/remove tuples from the dataset by assigning 0/1 to their variables, we need to consider that the output set has $2^5$ elements - one for representing the sum of each possible subset. Then, each of these elements would need to be associated with a unique value assignment, for example $\bar{a}_1 \cdot \bar{a}_2 \cdot \bar{a}_3 \cdot \bar{a}_4 \cdot \bar{a}_5$ for the first one, and $\bar{a}_1 \cdot \bar{a}_2 \cdot \bar{a}_3 \cdot \bar{a}_4 \cdot a_5$ for the second one, all the way to $a_1 \cdot a_2 \cdot a_3 \cdot a_4 \cdot a_5$ . Here we use the negation notation for variables which is out of the scope of our paper and we are just mentioning it here to illustrate an example. As we can see, using the provenance framework to reduce pipelines already includes an exponential explosion just to model the pipeline, and we are not even getting started with using it for computing the Shapley value. Hopefully this sheds light on the complexity induced by reduce operators and why we are so vigorously trying to avoid them.

---

> ### Author Response · Authors · 2023-11-22
> **Initial Comments - Part 3**
>
> (continued)
>
> (Q2) **Are modified KNN and quality metrics based on previous work or something new that the authors propose?**
>
> Firstly, just to clarify, the KNN model defined in section 4.1 is actually the vanilla KNN model. When we refer to that definition as "specific", we simply mean that it is slightly "alternative" because we split it into components that allow us to construct the counting oracle later on. Secondly, even though we are not specifically aware that someone else was using the definitions in the exact form defined in our paper, we would not count these definitions as major contributions. They are simply tools in our toolbox that we need for our actual contributions.
>
> (Q3) **What are the limitations of the work?**
>
> Since our work introduces various approximations, a key limitation stems from any situation where the approximation introduces a bias that negatively impacts the quality of the computed Shapley values and the resulting data debugging process. For example, there are cases when the KNN model does not accurately mimic the behavior of the target model on specific datasets. Also, sometimes the approximation defined in section 3.3 can introduce errors, for example, when the discrepancy between $\mathrm{reduce}(\mathcal{D})$ and $\mathrm{reduce}(\mathcal{D}\_{tr})$ is very large. Finally, sometimes the model quality metric may not be additive, for example, the F1 or R2 scores are not additive since the terms in the denominator do not depend only on $\mathcal{D}\_{val}$.
>
> (Q4) **Since we are using modified quality metric and ML algorithm (KNN), I wonder how practical is the approach in terms of non-KNN models and different quality metrics?**
>
> For clarity, we would like to point out that even though our method relies on approximating the target ML model using KNN, when evaluating our method, we are almost always applying it to target models that are not KNN. For example, in all figures in the main body of the paper, the target model is either logistic regression, XGBoost, or some deep learning model. One of the main goals of our evaluation section is to demonstrate that using the KNN model to approximate other models is indeed feasible and can lead to good results faster.

---

### Author Response · Authors · 2023-11-23

We are grateful to the reviewers for acknowledging the novelty, significance, and potential impact of our contributions, and we deeply appreciate all their constructive feedback. We did our very best to address all the concerns raised by them through our clarifying remarks posted below and extensive color-coded modifications to our paper draft. The opportunity to enhance our paper's quality was invaluable, and we thank the reviewers for their efforts. We hope that they will find that we managed to adequately address all the concerns they raised.

---

### Meta-Review · Area_Chair_Rkdz · 2023-12-08

**Metareview:**

This paper proposes a novel data debugging framework to identify influential training examples in a machine learning pipeline (including data preprocessing) via efficient Shapley value computation.

**STRENGTHS**

(1) The paper is thorough in introducing the new problem setting and motivating the significance of this problem.

(2) The proposed approach is technically sound with time complexity analysis.

(3) An extensive empirical evaluation is conducted to assess the performance of the proposed approach.


**WEAKNESSES**

(1) There were a fair number of questions/concerns regarding the clarity of presentation and some additional experiments. However, the authors have done a good job with addressing these concerns in great detail in their rebuttal as well as revising their paper to incorporate them.

(2) In the related work, there isn't much mention of Shapley value-based works in the context of data valuation which is highly related to data importance in this work. The authors can consider the following references (and the references therein) and discuss whether such works can be used within their pipeline:

Probably Approximate Shapley Fairness with Applications in Machine Learning. AAAI 2023.

Data Valuation in Machine Learning: "Ingredients", Strategies, and Open Challenges. IJCAI 2022.

https://github.com/daviddao/awesome-data-valuation

The authors are strongly encouraged to address the above concerns and the reviewers' in their revised paper.

**Justification For Why Not Higher Score:**

The use of Shapley value in quantifying data importance is not new even though the authors have used it in the generalized ML pipeline setting (in contrast to ML model).

**Justification For Why Not Lower Score:**

All the reviewers have acknowledged that this paper has sufficient technical merits for acceptance, as highlighted in the meta-review.

---

### Decision · Program_Chairs · 2024-01-16

Accept (poster)